

# Modulating surface heat flux through sea ice leads improves Arctic sea ice simulation in the coupled EC-Earth3

Tian Tian[1], Richard Davy[2], Leandro Ponsoni[3], and Shuting Yang[1]

[1]Danish Meteorological Institute, Copenhagen, Denmark
[2]Nansen Environmental and Remote Sensing Center, Bergen, Norway
[3]Flanders Marine Institute (VLIZ), Jacobsenstraat 1, Oostende, Belgium

**Correspondence:** Tian Tian (tian@dmi.dk)

**Abstract.** In this study, we address a persistent positive bias in Arctic sea ice (concentration and thickness) in the global climate model EC-Earth3 (ECE3) by including a modulating factor to the surface sensible heat flux over regions with sea ice concentrations above 70%, so-called ECE3L. We performed two pairs of 50-year simulations with repeated seasonal cycles: one pair replicating a cold climate and the other a warmer climate, with the latter characterised by thinner ice and weaker atmospheric boundary layer stability during winter. We show that modified heat flux can significantly alter surface air temperatures in the Arctic, with no substantial impact on lower latitudes. The changes are more pronounced in the cold climate, particularly during Arctic winter. We extended our comparison to two CMIP6 historical ensembles in a transient climate (1980-2014). We found that the mean sea ice states in the changing climate for the ECE3 (ECE3L) ensemble mean closely resembled the mean states in the cold-climate experiment. However, the reduction in sea ice area and volume achieved by ECE3L was nearly four times greater in the cold climate experiment than in the transient climate, reflecting the diminishing role of sea ice leads in a changing climate with decreasing occurrences of stable stratification in winter. Finally, our comparisons with satellite observations and reanalysis datasets demonstrated that ECEL3 significantly improves the local amplification ratio in the marginal ice zone of the Arctic, underscoring the importance of atmospheric stability shaped by central Arctic pack ice and its impact on Arctic amplification.

## 1 Introduction

Polar sea ice influences thermal interactions between the ocean and atmosphere by acting as an insulating barrier and reflective surface. The accelerated reduction of Arctic sea ice cover (extent and thickness) leads to increased solar absorption by the ocean (i.e. ice-albedo feedback mechanism), which in turn intensifies surface warming (Bhatt et al., 2014). As a result, Arctic warming rates have nearly quadrupled compared to the global average since satellite observations commenced in 1979 (Rantanen et al., 2022). The Arctic's rapid warming can increase the melting of Greenland ice sheet, raise global sea levels, extend and intensify Arctic fire seasons, speed-up permafrost thaw, and alter weather patterns in heavily populated mid-latitudes regions of the northern hemisphere (AMAP, 2021; Eyring et al., 2021; Thomas, 2017; Johannessen et al., 2020).

Accurate modelling of Arctic sea ice is essential for understanding and predicting the impact of climate change. However, several Earth System Models that contributed to the Coupled Model Intercomparison Project Phase 6 (CMIP6; Eyring et al.,



2016), including EC-Earth3, tend to simulate excessive sea ice in winter and an early minimum in August (Keen et al., 2021; Doescher et al., 2022) instead of September as indicated by observational, satellite-based datasets (Cavalieri et al., 1996; Stroeve et al., 2014; Fox-Kemper et al., 2021). In addition, most CMIP6 models struggle to reproduce the rapid decline in sea ice cover since the mid-2000s and a plausible evolution of Arctic warming at the same time (Notz and Community, 2020; Keen et al., 2021; Horvat, 2021). This persistent bias further undermines the reliability of future climate projections derived from
these models.

     Various efforts have been investigated in global climate models, with studies emphasizing the importance of reducing bias in the climate mean state during the historical period to better predict future sea ice changes (Massonnet et al., 2018; Docquier and Koenigk, 2021; Keen et al., 2021; Kay et al., 2022). Particularly in a warming climate, the thinning of sea ice and snow cover increases the importance of thermodynamic processes, involving heat and energy exchange between the sea ice, atmosphere,
and ocean surface (Massonnet et al., 2018; Landrum and Holland, 2022; Webster et al., 2021). Deser et al. (2010) demonstrated a connection between Arctic temperature inversion and sea ice loss, suggesting that the strong wintertime marine temperature inversion observed from 1980 to 1999 will diminish by 2080 to 2099. However, the presence of a positive bias in sea ice mean states can have profound consequences for the atmosphere, leading to unrealistic stable atmospheric stratification and ultimately damping the modelled sea ice sensitivity to external forcing.

In this study, we hypothesized that modulating upward heat flux through leads can remedy the seasonal bias in the coupled EC-Earth3 and improve the simulation of Arctic sea ice. This hypothesis is based on the understanding that the absence of parameterisation for turbulent exchange over leads in global climate models hampers adequately capturing the exchange of heat and energy between the atmosphere and the ocean through these crucial areas (Esau, 2007; Marcq and Weiss, 2012). Consequently, this deficiency may result in an early onset of stable stratification and an extended period of sea ice growth.

Implementing a new modulating factor to the Norwegian Earth System model, in its atmosphere-only model configuration (http://blueaction.eu/, Davy and Gao, 2019), significantly advanced our understanding of how heat fluxes through sea ice leads affect the Arctic's surface energy balance. This factor takes into account seasonal variations and varies based on the stability of the atmospheric boundary layer, increasing heat flux through leads to warm the air above during winter and dampening it during summer. Although the modulating factor shows promise in addressing known seasonal biases, its long-term climate
impacts remain uncertain due to potential changes in atmospheric stability and the spatial distribution of leads(Deser et al., 2010), highlighting the need for further investigation.

     To address these, we propose introducing the modulate factor to a coupled climate model. Specifically, this will allow us to investigate whether an amplified heat flux through sea ice leads in winter may accelerate the transition to a warmer Arctic with less perennial sea ice, potentially reducing the importance of leads under climate change. Moreover, we aim to assess whether
this modification can improve the sensitivity of climate models to external forcing in the Arctic. To do so, we will analyze changes in trends of key Essential Climate Variables (sea ice extent, area, volume, and surface air temperature) during a period of rapid Arctic change (1980-2014, Schweiger et al., 2019) due to the inclusion of the lead scheme. We will also identify the added value of this modification in reducing model bias on a regional scale.



## 2 Methods

### 2.1 The coupled EC-Earth3 with implementation of $A_{lead}$

In our study, we used a well-documented state-of-the-art global climate model EC-Earth3 (version v3.3) which is the model version contributed to the CMIP6 (Doescher et al., 2022). This model comprises three main components: atmosphere, ocean, and sea ice. The atmospheric component incorporates the Integrated Forecast System (IFS cycle 36r4) developed by the European Centre for Medium-Range Weather Forecasts (ECMWF), with a horizontal grid of TL255 and 91 vertical model levels. The ocean component uses the Nucleus for European Modelling of the Ocean, version 3.6 (NEMO3.6) embedded with the Louvain-la-neuve sea Ice Model, version 3 (LIM3, Rousset et al., 2015). The NEMO-LIM3 setup uses a nominal 1° resolution horizontal grid (i.e. ORCA1) and 75 vertical levels. In particular, the LIM3 sea ice model adopts an ice thickness distribution framework to deal with fine-scale ice thickness variations (Rousset et al., 2015).

EC-Earth3, hereafter referred to as ECE3, exhibits an Arctic sea ice bias in its mean state. While the total area of Arctic sea ice aligns well with satellite observations, there are generally large positive biases in the total volume of Arctic sea ice (Doescher et al., 2022), particularly compared to the Pan-Arctic Ice-Ocean Modeling and Assimilation System, often referred to as PIOMAS (Zhang and Rothrock, 2003), a reanalyses extensively validated (Stroeve et al., 2014; Wang et al., 2016) and broadly used by the community as a reference product (Davy and Outten, 2020; Keen et al., 2021). In September, the model shows an evident overestimation of Arctic sea ice thickness (SIT), with a bias of up to 2 m, while in March, ECE3 overestimates the central Arctic but underestimates it in the Bering and Kara Seas relative to PIOMAS (see Fig.13 in Doescher et al., 2022).

To alleviate the bias in the Arctic sea ice and assess its consequences on the global climate system, we introduced a factor $A_{lead}$ to the surface sensible heat flux (SSHF) within the coupled ECE3 framework, to better represent the heat exchange through leads in sea ice. Davy and Gao (2019, see Appendix A) outlined a method where the depth of the convective boundary layer ($\lambda_{CBL}$) is first determined by an empirical relation with the instantaneous input of air temperature gradient at a height of around 300 m. Using $\lambda_{CBL}$, the modulating factor $A_{lead}$ is further parameterised for $A_{max}$, considered as enhanced heat flux through narrow leads, namely when the fraction of open water is less than 10% (i.e. sea ice concentration SIC≥90%). On the other hand, when the open water fraction is greater than 30% (SIC<70%), the extra heat from narrow leads is assumed to be negligible and, therefore, $A_{lead}$ is treated as a constant of 1. Then, between 10-30% (i.e., 70%≤SIC<90%), it is linearly interpolated between 1 and $A_{max}$. The modulating factor $A_{lead}$ is applied globally on sea ice of both poles, updated every time-step with the instantaneous conditions of the lower-level air temperature gradients and sea ice concentration on the atmospheric model grid. The EC-Earth3 simulations with the implementation of $A_{lead}$ are hereafter referred to as ECE3L.

We emphasise that in ECE3L, the ocean does not supply additional heat to warm the atmosphere (Davy and Gao, 2019). Instead, its parameterisation modulates the efficiency of heat received by the surface atmosphere, depending on the open water fraction of the grid cell, under the condition of the same air-sea temperature differences. Specifically, when the sea ice leads are narrower, the atmosphere becomes more effective at receiving heat. This adjustment factor can increase to 1.2, enhancing surface heat exchange over sea ice where the sea ice concentration (SIC) is greater than 70%, particularly during





winter. Conversely, in summer, the modulating factor decreases from 1 to 0.9, resulting in the opposite effect on the surface atmosphere (see Fig. S1 and Section 3.2).

## 2.2 CMIP6 historical simulations and comparison strategies

We first performed a pairwise comparison of single simulations between ECE3L and ECE3 (i.e. with/without $A_{lead}$) in cold and warm climates (hereafter referred to as ExpCold and ExpWarm, respectively). Here we distinguished the warm climate scenario by its characteristics of thinner ice and weaker atmospheric boundary layer stability during winter compared to the cold climate (Fig. 1). To exemplify the contrasting conditions, we arbitrarily selected the years 1985 and 2015 to represent the warm and cold periods, respectively. The simulations used constant forcing with a repeating seasonal cycle corresponding to

the respective climate states, with initial states from one of the members of the ECE3 CMIP6 historical ensemble (see Fig. 3 by Doescher et al., 2022), specifically r5i1p1f1. We selected initial conditions from the historical simulation of r5i1p1f1 on January 1st in 1985 and 2015, respectively, and repeated external forcing for the respective years in a 50-year cycle. For each simulation, the first 20 years were designated as the spin-up period, with the subsequent 30 years for comparison purposes. We investigated whether implementing the new scheme in coupled climate simulations would result in any global impact, and if

this impact is influenced by the state of Arctic sea ice.

Next, we investigated how the presence of leads in Arctic sea ice affects the coupled climate system in a transient climate, particularly during significant Arctic warming due to climate change (1980-2014). Specifically, we focused on two key questions: 1) How might the importance of these open leads change during winter due to shifts in atmospheric stability? and 2) how do the differing sea ice evolutions in turn influence the modelled Arctic warming?

To address these, we performed a 20-member ensemble simulation using ECE3L and compared with the existing ECE3 CMIP6 ensemble (Doescher et al., 2022), both utilizing the same historical forcing of CMIP6. The ensemble size is determined by the availability of the ECE3 CMIP6 ensemble members, which are publicly accessible through the Earth System Grid Federation (ESGF, https://esgf.llnl.gov). These simulations were a subset of a 25-member ensemble (i.e. r1–r25i1p1f1 in Fig. 3 by Doescher et al., 2022), performed by the EC-Earth consortium following the CMIP6 protocol (Eyring et al., 2016) for

historical simulations of CMIP6 (1850-2014).

For ECE3L, we started with the initial conditions of two members of ECE3 in 1960 (i.e. r5i1p1f1 and r8i1p1f1). These two simulations, with $A_{lead}$, underwent a 15-year spin-up period. From 1975 onwards, we populated each of these simulations into 10 separate runs by means of perturbed atmospheric initial conditions, resulting in a total of 20 simulations with ECE3L, all running until 2014. The details of the experiments are summarized in Table 1.

## 120 2.3 Validation data and metrics

The model evaluation covers the period 1980-2014, aligning with the availability of satellite-based observational sea ice datasets in the Arctic. Our analysis involves comparing the two paired ensemble simulations. First, we calculate the bias, which was derived from the differences between the ensemble mean and the observed data. Second, we quantify the improve-





**Table 1.** Summary of pairwise experiments (ECE3 *versus* ECE3L)

| Pair | Experiment (model) Name | CMIP6 Forcing | # of Ensembles | Analysis Period |
|------|-------------------------|---------------|----------------|-----------------|
| 1 | ExpCold (ECE3 vs. ECE3L) | Repeated seasonal cycle for 1985 | 1 | 30 years |
| 2 | ExpWarm (ECE3 vs. ECE3L) | Repeated seasonal cycle for 2015 | 1 | 30 years |
| 3 | Ensemble (ECE3 vs. ECE3L) | Historical transient | 20[†] | 1980-2014 |

[†] ECE3 is a 25-member (r1–r25i1p1f1) ensemble (Doescher et al., 2022), with the realizations 6, 9, 11, 13 and 15 not publicly accessible.

ments by calculating the differences between the ECE3L and ECE3 ensemble means. These calculations enable us to evaluate
the model's ability to replicate observed conditions and to understand how the inclusion of $A_{lead}$ influences its performance.

For sea ice thickness, we relied on the PIOMAS reanalysis. Although PIOMAS is not strictly an observational dataset, it
is a valuable reference because it has been well validated against observations (e.g., Stroeve et al., 2014; Wang et al., 2016).
It also provides well-quantified measures of uncertainty (Schweiger et al., 2011) and is commonly used to evaluate climate
models (Davy and Outten, 2020; Keen et al., 2021). For sea ice concentration, we used two independent observational datasets.
One referred to as NSIDC-0051, which is derived from passive microwave data and has been processed using the NASA
Team algorithm (Cavalieri et al., 1996). The other referred to as OSI-450a, known as the global sea ice concentration climate
data record, version 3.0 (2022). It was sourced from OSI SAF (Ocean and Sea Ice Satellite Application Facility). Given that
PIOMAS assimilates sea ice concentration data from NSIDC products, we considered NSIDC-0051 as the primary reference.
The data from OSI-450a served as a second reference. We performed assessments using both datasets, with additional details
available in Supplements.

For model evaluation, we maintained consistency by regridding all sea ice data, from both climate models and observations,
to the NSIDC-0051 polar stereographic grid with a 25 km spatial resolution, following Lin et al. (2021). We calculated sea
ice area, extent (SIC>15%) and volume (i.e. mulitiplying sea ice area by the sea ice thickness) across the entire Northern
Hemisphere ice-covered region using monthly mean data from 1980 to 2014. To evaluate the accuracy of modelled sea ice
edges compared to observations, we used the Integrated Ice Edge Error (IIEE) metric introduced by Goessling et al. (2016),
applying a criterion of SIC=15% to define the sea ice edge. The IIEE quantifies the total area where the modeled SIC differs
by more than 15% from the reference data. It accounts for regions where the model either overestimates (O) or underestimates
(U) SIC relative to the 15% threshold. In summary, the IIEE is calculated as the sum of these areas: IIEE = O + U. This metric
offers valuable insights into how well the modelled sea ice edges align with observational references (e.g., Ponsoni et al.,
2023).

To evaluate how changes in surface heat flux can influence temperature patterns and trends in the Arctic and globally, we
selected four global surface temperature datasets as reference fields. These datasets include ERA5 (Hersbach et al., 2020),
NCEP2 (Kanamitsu et al., 2002), and JRA-55 (Kobayashi et al., 2015), which are three atmospheric reanalysis datasets provid-
ing air temperature at 2 meters (T2m). Additionally, we use the Berkeley Earth land/ocean temperature dataset (BEST, Rohde
and Hausfather, 2020), which is different from reanalysis data sets as it combines its own land surface temperature records
with air temperature data and utilizes the HadSST4 dataset for sea surface temperatures (SSTs, Titchner and Rayner, 2014).





Following Rantanen et al. (2022), we calculate Arctic amplification (AA), defined as the ratio of Arctic warming to global mean warming. The Arctic region is defined as the area encircled by the Arctic Circle (66.5°–90°N). The slopes of linear trends in surface temperatures used least-squares fitting for the annual mean values in both the Arctic and the global domains. In this study, our main objective is to evaluate how amplified heat flux reduces the overestimation of modelled sea ice in the Arctic. Therefore, we do not include an analysis on the underestimated sea ice in the Antarctic.

## 3 Cold *versus* warm climate

### 3.1 Atmospheric stability and sea ice variability

The turbulent processes over sea ice are affected by the temperature difference between the air and the ice surface, which is influenced by sea ice concentration (lead cover) and ice thickness (Lüpkes et al., 2008). In the ECE3 (baseline) simulations, the occurrence of the low-level winter temperature inversion is defined as positive air temperature differences between 850 hPa and 1000 hPa, following the method of Deser et al. (2010). In Figure 1a, winter temperature inversion over Arctic sea ice (SIC≥ 70%) is considerably weaker and shorter in ExpWarm compared to ExpCold on a 30-year average. Particularly during the early freeze-up months (October and November), mean differences in a warmer climate are nearly half as small as in a colder climate, with the central Arctic pack ice (SIC≥ 70%) experiencing over 1 m of thinning and more than 2 million km$^2$ of shrinkage during summer (Figs. 1b,c) compared to ExpCold. The comparison between two baseline simulations suggests that the effect of parameterising turbulent process on sea ice becomes less pronounced during Arctic warming, likely due to a reduction in the strength and duration of winter temperature inversion.

In the northern hemisphere, the 30-year mean sea ice area (SIA) and volume (SIV) in the ECE3 simulations for ExpWarm account for 84 % and 57 % of those for ExpCold, respectively (Figure 2). It is equivalent to a decrease in the mean thickness, defined as the total SIV divided by SIA, from 2.7 to 1.8 m. In the ECE3L simulations, the 30-year mean changes caused by modified heat fluxes through lead are less pronounced in ExpWarm than in ExpCold (Fig. 2). Specifically, SIA is 2% (12 %) less in ECE3L than in ECE3, while SIV is 7% (27 %) less in ECE3L during the 30-year time-frame for ExpWarm (ExpCold). The results support our hypothesis that the model's response to the modulation of surface heat flux can be influenced by the winter temperature inversion and the extent of thick ice (Fig. 1). Furthermore, modulating surface heat flux over sea ice results in persistent reduction of sea ice area and volume in ExpCold for each simulation year, while it can either increase or decrease sea ice in ExpWarm during the 50-year cycle (as revealed in the full time series in Fig. S2). In ExpWarm, we observe that in years when ECE3 exhibits a mean thickness of 1.6 m (1.9 m), ECE3L tends to increase (decrease) sea ice.

Globally, we find that the 30-year mean of global surface temperature in ExpCold is 0.21°C higher in ECE3L than ECE3, which falls within the range of model variability across the ECE3 20 members of historical simulations (which is 0.24° for the respective year). Similarly, the change is only 0.02°C higher between ECE3L and ECE3 in ExpWarm, relative to the model spread of 0.18°C. This indicates that the change in global mean surface temperature induced by modulation of heat flux over sea ice in ECE3L is within the range of natural variability represented by the model ensemble. In particular, the modification does not cause global surface temperature drift, and both models exhibit robust internal variability, as illustrated in Figure S3.





## 3.2 Local and remote influences of sea ice leads

The region with thick ice (SIT>2 m) in winter is considerably smaller in ExpWarm compared to ExpCold (Figures 3a, c). Consequently, the thinning of sea ice differs between these two climate scenarios. In ExpCold, it reaches a maximum of -1 m in the western central Arctic, while in ExpWarm, the maximum thinning is -0.5 m in the eastern central Arctic (as shown in Figs. 3b,d). This shift from the eastern Arctic to the western Arctic coincides with regions experiencing peak winter amplification of heat flux through leads (refer to Figs. S1a, c). This spatial pattern is governed by the prevailing conditions of atmospheric instability. In Figures 4a,c, the sea ice cover shows similarities between ExpCold and ExpWarm in ECE3L. However, when compared to ECE3, there is a considerable reduction in sea ice concentration (where SIC<70%) in the North Atlantic marginal ice zone in ExpCold, whereas there are only little changes in the Greenland Sea in ExpWarm, as illustrated in Figures 4b,d. The findings suggest that during Arctic winters, the decrease in sea ice concentration is mainly due to heat advected by the ocean from the south, rather than air-sea heat exchange through sea ice leads.

In Arctic summer, the areas experiencing maximum thinning of sea ice remain consistent with those observed in winter (as shown in Fig. S4). However, in ECE3L, sea ice is slightly thicker in the marginal seas surrounding the Arctic Ocean in both climate scenarios. Additionally, there is a notable reduction in sea ice concentration up to 30% in ExpCold compared to less than 20% in ExpWarm, as illustrated in Figure S5. These results clearly show a reduction in the magnitude of heat flux amplification, corresponding to a decline in the mean states of sea ice as the climate shifts from colder to warmer conditions, observed in both winter and summer months.

Statistically significant differences in surface air temperature between ECE3L and ECE3 are noted at high latitudes in all seasons in ExpCold (Fig. 5), unlike in ExpWarm where the differences are minor. In regions with less sea ice in ECE3L compared to ECE3, the surface is warmer in ECE3L, especially in non-summer seasons. These findings underscore the role of sea ice in shaping polar surface temperatures. A warmer atmosphere can facilitate greater moisture convergence, increasing precipitation, especially in the Arctic. Therefore, variations in precipitation generally reflect the temperature differences between ECE3L and ECE3, with the magnitude being negligible (not shown), aligning with the findings of Kay et al. (2022). Regarding polar sea level pressure, differences are modest except for a significant drop during Arctic winter in ExpCold (Fig. 6), where ECE3L's warmer boundary layer causes a low-pressure response, indicative of the typical baroclinic structure in atmospheric circulation due to thermal forcing (Deser et al., 2010).

## 4 Transient climate: comparison of CMIP6 20-member ensemble simulations

### 4.1 Enhanced performance: reduced seasonal bias and narrower model spread

The ensemble mean of ECE3L consistently shows lower sea ice area and volume than that of ECE3 throughout the annual cycle, with the largest differences noted during winter when ECE3 reaches its peak (Fig. 7). ECE3L more closely aligns with the observed seasonal cycle than ECE3. However, during the summer months (June to August), both models slightly underestimate sea ice area, with ECE3L showing a slightly larger mean difference of -0.6 million km$^2$ compared to -0.4 million km$^2$ for ECE3



in June. This underestimation aligns with a common bias across several coupled CMIP6 models, including ECE3, where the minimum summer Arctic Sea Ice Area occurs in August rather than September (Keen et al., 2021; Doescher et al., 2022). Consequently, the largest biases are observed in September, amounting to 0.7 million $km^2$ for ECE3L and 1 million $km^2$ for

ECE3. Both models consistently overestimate Arctic sea ice volume throughout the year, with positive monthly mean biases ranging from 6 to 11.2 thousand $km^3$ in ECE3L and from 7.8 to 12.9 thousand $km^3$ in ECE3. These biases are reduced year-round by the lead scheme, however, they are not significantly different from those in the ECE3 historical ensemble. The ensemble spreads range from 2.6 to 3.2 thousand $km^3$ in ECE3L and from 4.8 to 5.5 thousand $km^3$ in ECE3.

Further comparisons highlight that the September Arctic sea ice extent in the ECE3L ensemble closely matches observations

from 1980–2014, outperforming ECE3 (Fig. 8b). Across most years, ECE3L consistently shows less September sea ice extent than ECE3 (Fig. 9a). In March, the Arctic sea ice volume in ECE3L shows a closer alignment with the PIOMAS reanalysis, demonstrating its improved performance over ECE3 (Figs. 8c,d and 9b). However, despite these improvements, both models do not fully capture the observed trends in sea ice decline. In September, the models underestimate the declining trend in sea ice extent, showing a decrease of -0.6 million $km^2$ per decade compared to the observed -0.9 million $km^2$ per decade. Similarly, in

March, both models overestimate the declining trend in sea ice volume, with trends of -3.4 thousand $km^3$ per decade for ECE3 and -3.1 thousand $km^3$ for ECE3L, against an observed -2.5 thousand $km^3$ from PIOMAS.

The model spread of ECE3 exhibits notable decadal changes before and after the 1990s in the transient climate, whereas the model spread of ECE3L remains relatively consistent and smaller compared to ECE3 (Figs. 8 and 9). Before 1990, ECE3L achieved substantial reductions in the overestimation of sea ice area and volume. The improvements suggest that incorporating

the amplification effect through leads in the central pack ice can refine estimates of the declining sea ice trend, at least to some extent. However, the effectiveness of this approach is diminishing due to the evolving Arctic climate, characterised by decreased occurrence of stable stratification in winter (Deser et al., 2010).

### 4.2 Enhanced Performance: reduced regional bias and improved representation of sea ice edge

Arctic sea ice thickness fields reveal the impacts of heat flux modulation, during the March maximum and September minimum

periods (Fig. 10). Comparisons between the ensemble means show that in March, the sea ice thickness in transient climate for ECE3L, generally exceeding 3 m in the central Arctic, is greater than the 30-year mean for ExpCold (Figs. 3a,b). However, the mean difference in ice thickness between ECE3L and ECE3 (i.e., thinning of central pack ice where SIC≥70%) is considerably smaller than the changes observed in ExpCold (Figs. 3a,b). Similarly, in September, the central Arctic sea ice thickness in the transient climate for ECEL3 is substantially greater than during ExpCold, although the extent of these changes (ECE3L-ECE3)

is smaller (Figs. S4a,b).

Arctic sea ice concentration maps for March and September climatologies in both ensemble means (Fig. S6) closely resemble those observed in the mean states for ExpCold (Figs. 4a,b and S5a,b), but with moderate differences between ECE3L and ECE3. To accurately assess these models, we employ the integrated ice edge error (IIEE) metric and compare the results with satellite observations from NSIDC (Fig. 11). As documented by Doescher et al. (2022), in March, ECE3 tends to overestimate ice

concentration near the ice margins in the Atlantic sector, while underestimating it in the Bering Sea and Sea of Okhotsk (named





in Fig. 11d). ECE3L shows a notable improvement in reducing the positive bias found in the Atlantic Sector, though it slightly increases the negative bias in the Pacific Sector, noted in the Sea of Okhotsk. In September, ECE3 generally overestimates sea ice concentration at ice margins, except in the Kara Sea, where it underestimates. The improvement by ECE3L is particularly noticeable in the Atlantic sector. These findings are corroborated by using an alternative satellite dataset (Figs. S7 and S8), supporting the conclusions drawn by Doescher et al. (2022). The monthly time series of IIEE indicate that ECE3L consistently outperforms ECE3 throughout the period (Fig. S9). Notably, the model spread of ECE3, especially in winter months (Fig. 12), has dramatically decreased since the 2000s. In contrast, the model spread of ECE3L has remained relatively low, with no apparent decadal shifts.

In summary, the annual climatologies for the changing sea ice conditions in the transient climate, as represented by the ECE3 and ECE3L ensemble means (Fig. 7), closely match those of the repeated climate scenario for ExpCold (Fig. 2). For the period 1980-2014, the sea ice area and volume were 10.4 million $km^2$ and 27.6 thousand $km^3$, respectively, representing a 3% and 7% reduction compared to ExpCold, with sea ice area and volume of 10.9 million $km^2$ and 29.3 thousand $km^3$, respectively, showing reductions of 12% and 27%. Notably, the mean sea ice thickness (defined as SIV devided by SIA) in both climate conditions for ECE3 is 2.7 m, yet the reductions in area and volume by ECE3L are nearly four times greater in the repeated climate for ExpCold. This underscores the diminishing influence of sea ice leads in modifying the Arctic climate, largely due to reduced occurrences of stable stratification in the winter as the Arctic warms.

## 5 Discussion

### 5.1 Implication of the differing sea ice evolution for Arctic warming

This study has substantial implications for Arctic climate modeling. By incorporating a modulating factor for surface sensible heat flux over sea ice to account for the processes over leads, the EC-Earth3 model has significantly improved its accuracy in simulating Arctic sea ice extent and volume. This critical adjustment alleviates a persistent bias reported in previous simulations. Such a model development is essential for understanding the Arctic's response to climate change and for enhancing the reliability and predictive capabilities of global climate models. Recent research emphasizes the importance of accurately representing sea ice processes to capture the complex feedback mechanisms that drive Arctic amplification and impact global climate patterns (AMAP, 2021; Docquier and Koenigk, 2021; Kay et al., 2022). The improved model performance described in this study supports these insights, highlighting the need for parameterisations of small-scale ocean-sea ice-atmosphere coupling processes to improve predictions of future Arctic conditions.

Focusing on surface warming, a primary indicator of climatic impacts, this section evaluates the differing impacts of sea ice evolution modeled by ECE3 and ECE3L on climate change. Specifically, it explores how these variations affect regional warming patterns, thereby enhancing our insight of both localized and broader implications of Arctic warming. Given the challenges in accurately measuring absolute air surface temperatures in the Arctic and globally, it remains uncertain whether ECE3L provides a better representation of the mean state and the warming trend, as highlighted by Rantanen et al. (2022) and Tian et al. (2024). To address this, we adopt Rantanen et al. (2022)'s method and compare anomalies against the average of four





observational or reanalysis datasets. This approach, as highlighted by Tian et al. (2024), combines observational data to mini-
mize the artifact effects of warm bias over the Arctic sea ice in the reanalyses under very cold conditions. In the period 1980 to
2014, the annual mean temperature anomalies within the Arctic region (66.5°–90°N) show slight discrepancies between ECE3
and ECE3L only before 1995 (see Fig. S10a). The global mean temperature anomalies present indistinguishable differences
between the two models throughout the entire period (see Fig. S10b). It is worth highlighting that both models resemble the
evolution of the observed Arctic air surface temperature anomaly, but consistently overestimate the global warming trend from
1980 onwards (Doescher et al., 2022). This discrepancy ultimately results in an underestimated Arctic amplification ratio.

The temperature trend maps for 1980-2014 (Fig. 13a-c) demonstrate that ECE3L outperforms ECE3, especially in the
marginal ice zone. ECE3 significantly overestimates the warming trend in the Barents Sea, whereas ECE3L provides a more
accurate representation, extending the trend southwestward to the Greenland and Labrador seas, more closely aligning with
observed trends. Additionally, ECE3L extends the warming trend into the East Siberian Sea, unlike ECE3, which consistently
underestimates the trend in the eastern Arctic (specific locations given in Fig. S1d). These improvements in ECE3L result in
a more accurate local amplification ratio (Fig. 13d-f), primarily due to differing sea ice concentration evolution patterns in the
Arctic (Fig. S7b,c). Such regional benefits further improve the accuracy of Arctic amplification estimates, specifically within
the region 66.5–90°N (Rantanen et al., 2022).

## 5.2 Advances and limitations

Current climate models often exhibit significant seasonal biases in sea ice simulations, which compromise the accuracy of long-
term climate projections for the Arctic (Doescher et al., 2022; Keen et al., 2021). Studies by Deser et al. (2010) and Frankignoul
and Kwonb (2024) highlight how misrepresentations of critical feedback mechanisms, such as the ice-albedo feedback, can
lead to inaccuracies in seasonal sea ice predictions. This feedback is crucial in Arctic amplification, where retreating sea-ice
leads to more open water, which absorbs more solar radiation, further warming the region and exacerbating sea ice melt in a
cycle that accelerates the decline of Arctic sea ice. These biases not only impact the Arctic atmospheric stability but also affect
global weather patterns.

To improve the accuracy of seasonal predictions and climate projections for the Arctic, it is essential to account for unre-
solved oceanic and atmospheric coupling processes (Eyring et al., 2021; Docquier and Koenigk, 2021). Researches (Marcq
and Weiss, 2012; Esau, 2007) have emphasized the significant impacts of turbulent heat exchanges over leads and the need
for higher resolution in modeling these processes. However, sea ice leads are fractal in nature on scales of a few meters up to
tens of kilometers. This is an important source of uncertainty as the fractal dimension of number of leads of a given size likely
depends upon properties of the sea ice floe such as thickness, damage, and age. However, these relationships have not yet been
found, and so Davy and Gao (2019) employed a fixed fractal dimension derived from satellite observations. Furthermore, how
the fluxes from leads depend upon lead width, background wind, and orientation is the subject of ongoing research using turbu-
lence resolving models for the atmospheric boundary layer (Esau, 2007; Gryschka et al., 2023). The results of Davy and Gao
(2019) that were used to create the modulation factor employed here depended on the scale sensitivity derived from the model
results of Esau (2007). However, recent work by Gryschka et al. (2023) has shown that the dependency of fluxes on lead width





can be very different depending on the roughness of the surrounding sea ice. This uncertainty from the turbulence resolving modelling highlights the need for further research in this direction to reduce the model-form uncertainty. Given the large differ-

320 ences found in different turbulence resolving model results it is also crucial to constrain these model results using observations from leads of a wide range of widths. This could be achieved using recent developments in drone-based measurements.

Our study advances the field by directly modulating heat flux through sea ice leads with a coupled climate model, a topic not extensively explored in previous studies (Davy and Gao, 2019). This modification significantly improves the model's ability to dynamically adjust (either increase or decrease) sea ice states based on local sea ice conditions and the prevailing atmospheric

conditions. Comparative analysis shows that ECE3L more accurately captures Arctic sea ice variability than ECE3, aligning more closely with observational data from PIOMAS and NSIDC (Schweiger et al., 2011; Stroeve et al., 2014), and effectively minimizing uncertainties (i.e. reduced model spread) in Arctic sea ice simulations. This improvement is particularly noted under conditions of thicker ice in the central Arctic and stronger winter atmospheric stability. Additionally, our modifications to sea ice melt processes show limited global implications, consistent with findings from Kay et al. (2022), highlighting the

potential for refining predictions of sea ice dynamics and their climatic impacts.

Despite these advancements in modeling Arctic climate, significant challenges remain. Current models, including EC-Earth3, often fail to accurately capture the accelerated decline of Arctic sea ice volume (Keen et al., 2021). Massonnet et al. (2018) have emphasized that the mean state of sea ice, especially its thickness, is crucial for activating key feedback mechanisms that enhance model sensitivity to external forcing. These feedbacks are vital for reliably predicting the timing and

335 implications of an ice-free summer in the Arctic Ocean. Additionally, ECE3L consistently overestimates sea ice volume up to the end of the analysis period, associated with the diminishing influence of sea ice leads, largely due to the weakening of winter stability. Addressing these deficiencies requires further research, particularly into how variations in sea ice and snow thickness affect heat and energy exchanges in the Arctic (Landrum and Holland, 2022). It is also essential to investigate how different representations of these processes can alter climate model sensitivity to external forcing (Webster et al., 2018).

Furthermore, this study relies on specific model configurations and parameterisations, which may not be directly applicable across different climate models. As indicated by Chen et al. (2023), significant inter-model spread in Arctic sea ice thickness within CMIP6 simulations compared to PIOMAS data highlight the need for broader application tests. Exploring the adaptability of the modulating factor approach across different models could help validate and generalize the findings, ultimately enhancing global climate projections.

The modulation factor $A_{lead}$ is applied globally, including the Antarctic. However, its local effect is confined to sea ice in the Weddell Sea and Ross Sea in ECE3L (not shown), due to a substantial warm bias in the Southern Ocean and the resulting underestimation of Antarctic sea ice, as identified in the ECE3 CMIP6 historical simulations (see Figs. 10 and 14 of Doescher et al., 2022). This diminishes the effect of parameterisation in the Antarctic. Further refining parameterisation to account for thinner sea ice in both the Antarctic and warming Arctic could provide insights into the contrasting behaviors and feedback

mechanisms of sea ice, enhancing our understanding of polar climate interactions and potentially improving the accuracy of global climate projections.



# 6 Conclusions

This study showcases that the persistent positive bias in Arctic sea ice simulated within the global climate model EC-Earth3 can be effectively alleviated by introducing a modulating factor, which adjusts surface sensible heat flux through leads in the central pack ice. Our comprehensive evaluation, involving two sets of 50-year simulations and comparisons with CMIP6 historical ensembles, demonstrates that the modifications significantly affect winter surface air temperatures in the Arctic, while having minimal impact on non-polar regions.

The spatial changes in sea ice mean states from 1980 to 2014, induced by model modifications in ECE3L, closely mirror those simulated under the colder climate conditions of the 1980s. However, the reduction in total Arctic sea ice area and volume is nearly fourfold greater in the 1980's, a period characterised by stronger atmospheric stability. This implies a diminished effectiveness of sea ice leads in modulating air-sea heat exchange, a result of ongoing changes in winter Arctic atmospheric stability. The enhanced model agreement with observed and reanalysis data, particularly in the North Atlantic marginal ice zone, emphasizes the critical role of atmospheric stability in shaping both the state of sea ice and the broader patterns of Arctic amplification.

In a warmer climate, the modulating factor can either increase or decrease sea ice states depending on the prevailing atmospheric stability and mean sea ice thickness, highlighting its potential to improve the predictive capabilities of global climate models. It underscores the importance of accurately simulating sea ice dynamics to better understand Arctic climate response to forcing. The next step is to refine the parameterisation to include an ice thickness-dependent modulation factor, effectively incorporating both Antarctic and Arctic sea ice simulations in a warming climate. This will help explore their broader implications for global climate projections and inform effective strategies for mitigating the impacts of climate change.

*Code and data availability.* The model output from EC-Earth3 CMIP6 historical simulations are available freely and publicly from the Earth System Grid Federation (ESGF, https://esgf.llnl.gov). There are in total 20 members available and hence used here, namely r1-5, 7,8,10,12,14,16-25. The PIOMAS monthly outputs are available at http://psc.apl.uw.edu/research/projects/arctic-sea-ice-volume-anomaly/data/ (last accessed on 8 June 2024). The global sea ice concentration climate data record OSI-450a used here can be accessed at https://osi-saf.eumetsat.int/products/osi-450-a (last accessed on 8 June 2024). The diagnostic package for the analysis of NEMO model output CDFTOOL (v3) is available at https://github.com/meom-group/CDFTOOLS (last access on 8 June 2024).

*Author contributions.* RD developed the algorithms. TT, RD, and SY conceived the idea, implemented the algorithms to the EC-Earth3 model, and designed the experiments. TT carried them out and analyzed the results. LP collected satellite-based observational sea ice datasets. TT and LP performed sea ice validation. TT prepared the manuscript with contributions from all co-authors and all authors discussed the results at all stages.



*Competing interests.* The authors declare no competing interests.

*Acknowledgements.* TT, RD and SY have been supported by the Blue-Action project (European Union's Horizon 2020 research and innovation program, no. 727852). TT and SY were supported by the Danish National Center for Climate Research (NCKF).



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






**Figure 1.** Seasonal air temperature differences between the 850hPa and 1000hPa levels over Arctic sea ice compared between ExpCold (dark blue) and ExpWarm (light gray) scenarios, based on the ECE3 baseline simulations (a). Error bars indicate one standard deviation of the 30-year variability. The temperature data is area-averaged for regions with sea ice concentration (SIC) ≥70% in the Arctic Ocean (66.5–90°N). Additionally, the 30-year mean differences in sea ice thickness between ExpWarm and ExpCold are shown on the right y-axis, in solid circles (SIC≥70%) and in asterisks (SIC≥80%). Panels (b) and (c) illustrate air temperature differences in October on 30-year averages in ExpCold and ExpWarm, respectively. The black thick line indicates the Arctic Circle (66.5°N), within which sea ice area for SIC ≥70% is calculated in (a) for each month.



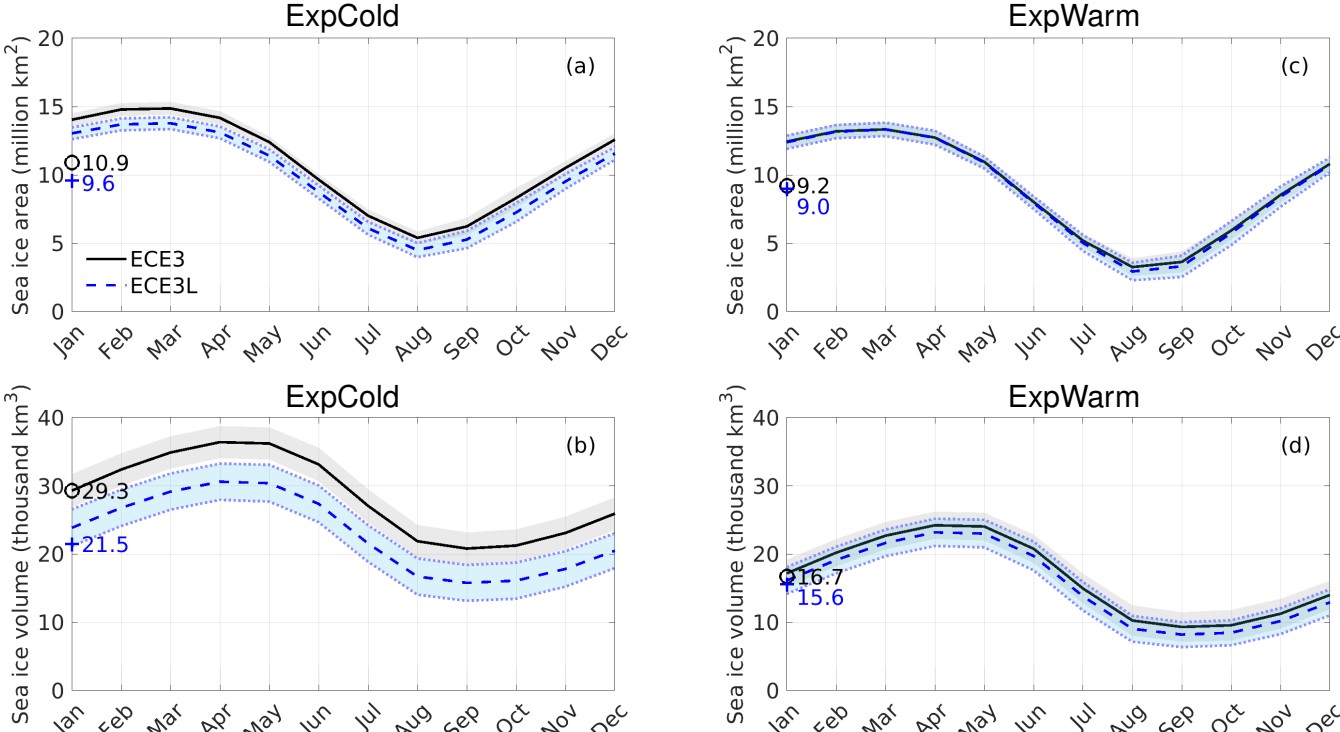

**Figure 2.** Seasonal cycle in ECE3 (black solid line) and ECE3L (blue dashed line) over the Arctic under a constant forcing: (a) Sea ice area and (b) Sea ice volume. Simulations are performed for ExpCold, characterised by thicker ice and stronger stability of the atmospheric boundary layer during winter in the Arctic Ocean. Values are shown as the mean (thick line) and one standard deviation (shaded area) over the last 30 years. The full time series of simulations is shown in Fig. S2. (c) and (d) as in (a) and (b), but for ExpWarm with thinner ice and weaker static stability. SIE (SIA) and SIV are calculated with the cdficediags tool (from CDFTOOLS 3.0). Numbers and symbols indicate the 30-year mean for respective variable for ECE3 (black open circles) and ECE3L (blue crosses).



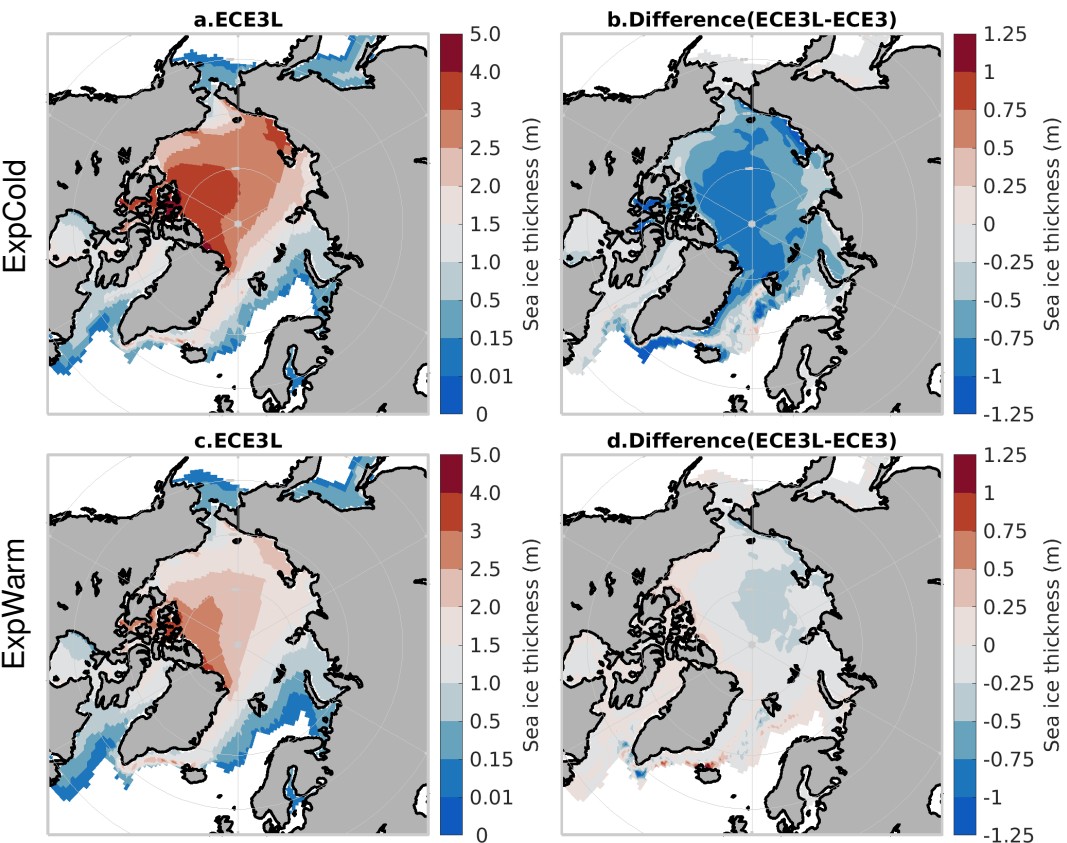

**Figure 3.** Late winter (March) Arctic sea ice thickness under a constant forcing: for ExpCold (a) ECE3L, (b) Differences (ECE3L-ECE3). (c) and (d) as in (a) and (b), but for ExpWarm. Values are shown as 30-year averages as in Fig. 2. Note: nonlinear color scale is used in (a) and (c) to emphasize thin ice categories. Thickness under 0.01 m is not shown. Note that the PIOMAS domain is defined as SIT>0.15 m (see colorbar). Late summer Arctic is shown in Fig. S4.



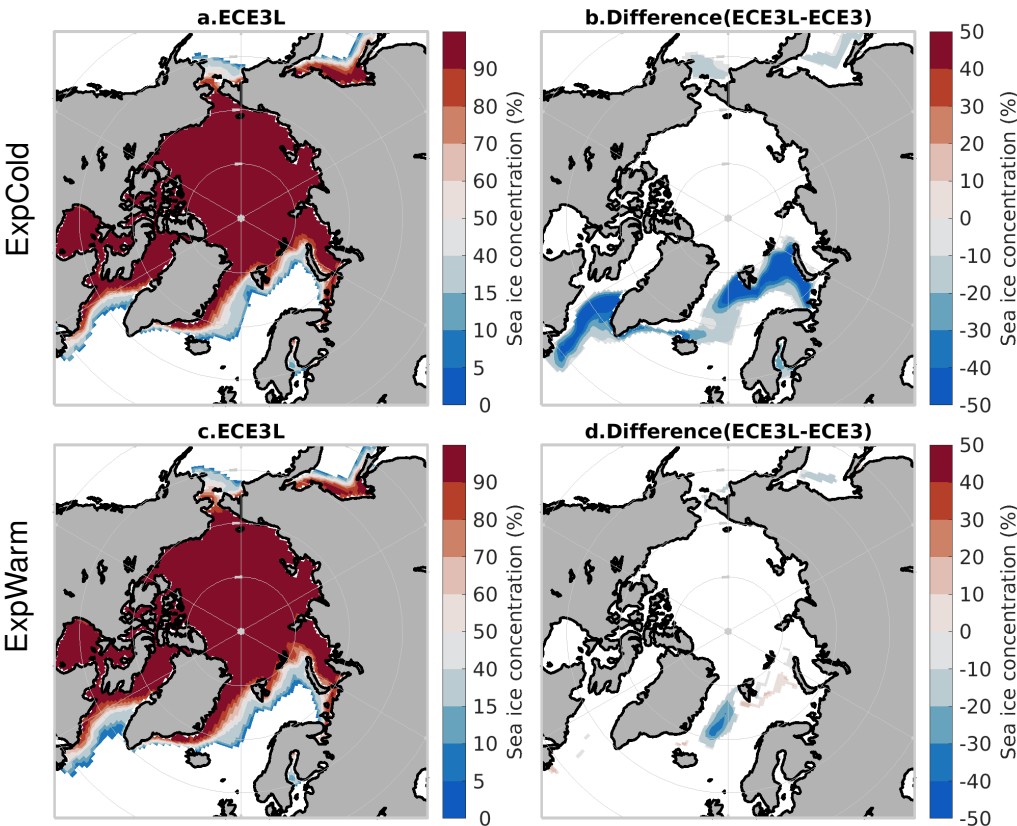

**Figure 4.** Late winter (March) Arctic sea ice concentration under a constant forcing: for ExpCold (a) ECE3L, (b) Differences (ECE3L-ECE3). (c) and (d) as in (a) and (b), but for ExpWarm. Values are shown as 30 year averages as in Fig. 2. Note: nonlinear color scale is used to emphasize low ice concentration. Concentration under 5 % is not shown. Late summer Arctic is shown in Fig. S5.





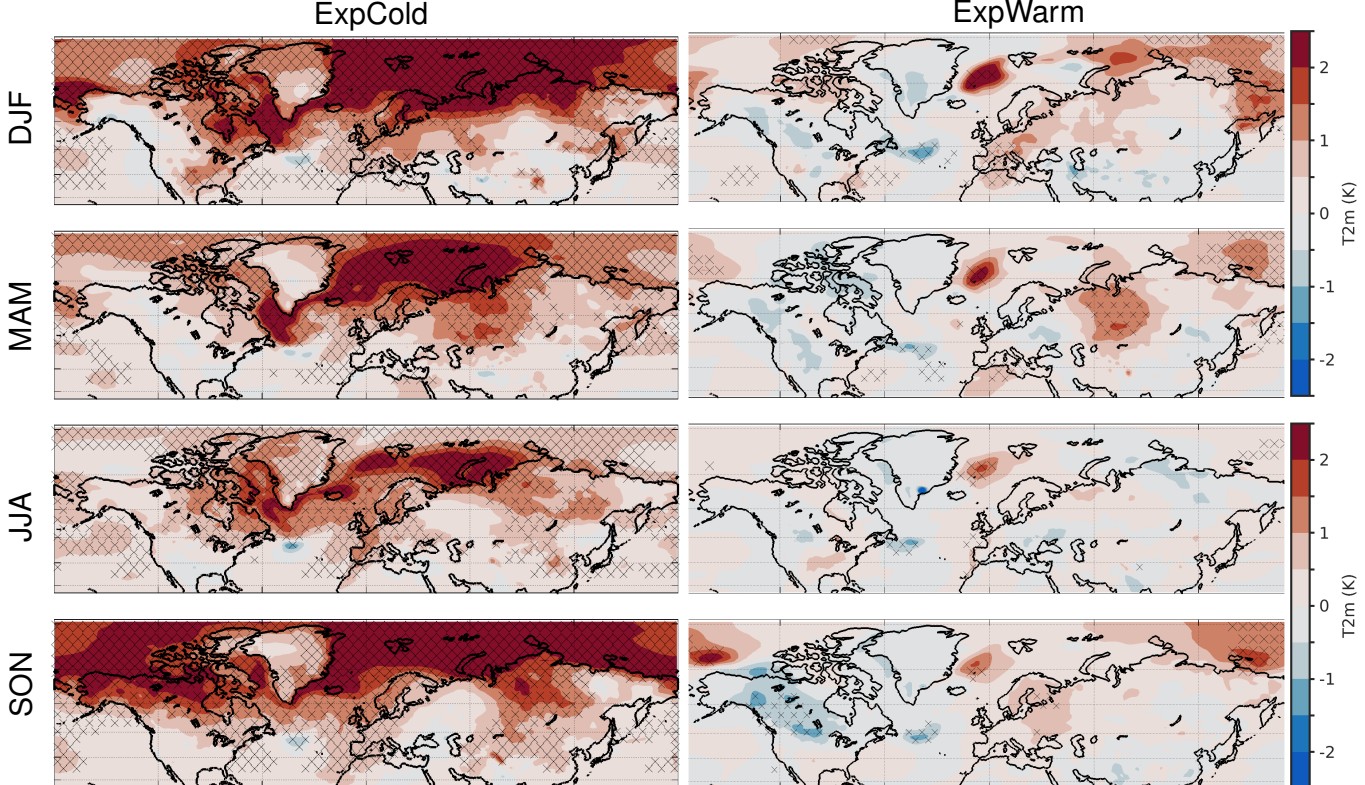

**Figure 5.** Seasonal differences in 2m temperature (K) between ECE3L and ECE3 in the northern hemisphere ($20°–90°$N) for ExpCold (left) and ExpWarm (right). Stippling indicates areas with statistically significant differences ($p < 0.05$) determined by a two-sided $t$-test.





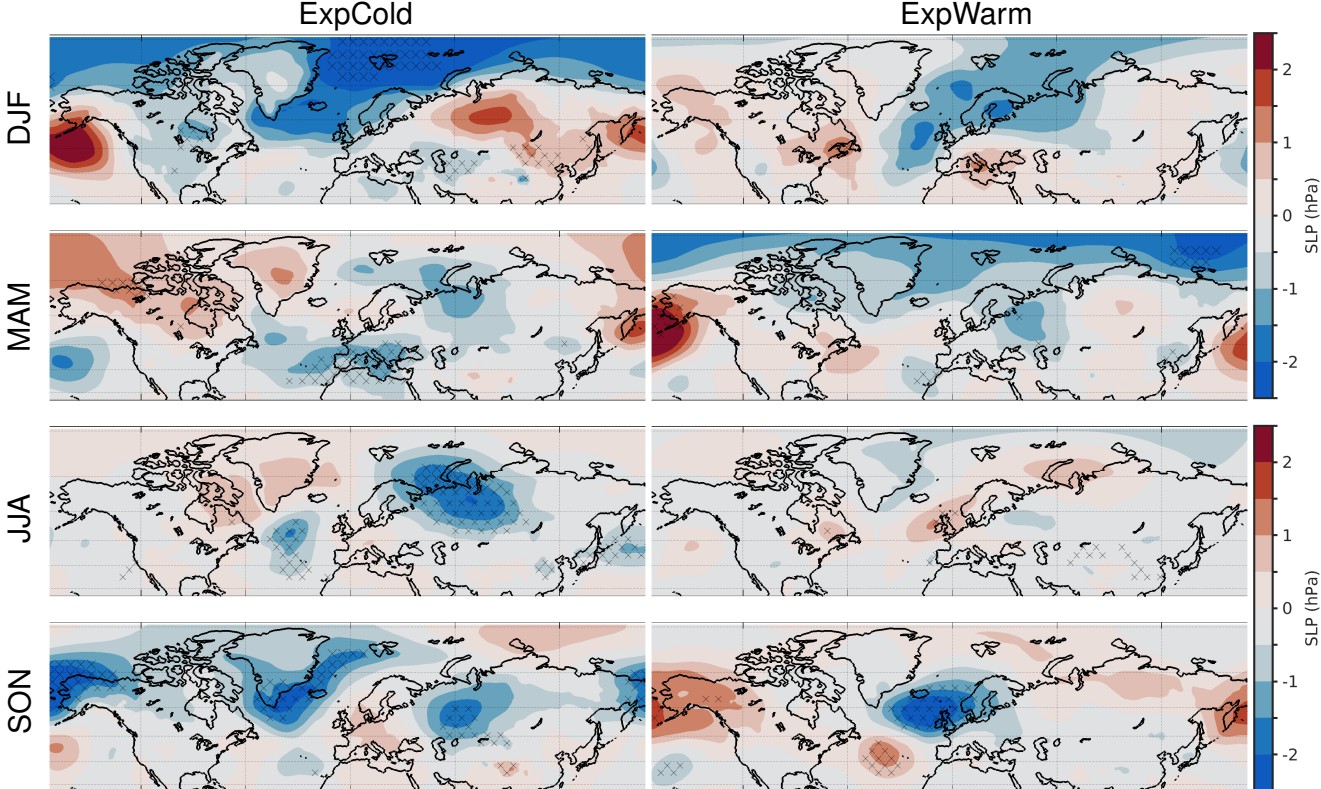

**Figure 6.** As in Figure 5, but for SLP (hPa).





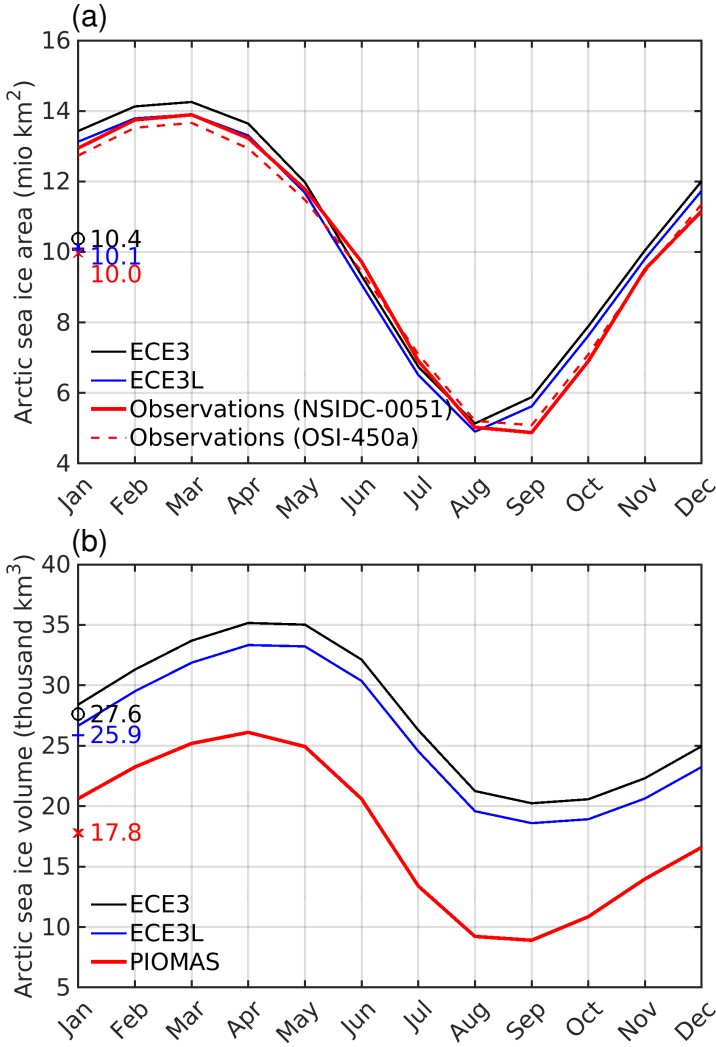

**Figure 7.** Comparison of seasonal cycle in the transient climate (1980–2014) between ECE3 and ECE3L for (a) Arctic sea ice area and (b) Arctic sea ice volume. Observations for the sea ice area include NSIDC and OSI-450a datasets, both remapped to the NSIDC-0051 grid. Sea ice volume is based on PIOMAS domain criteria (thickness > 0.15 m).



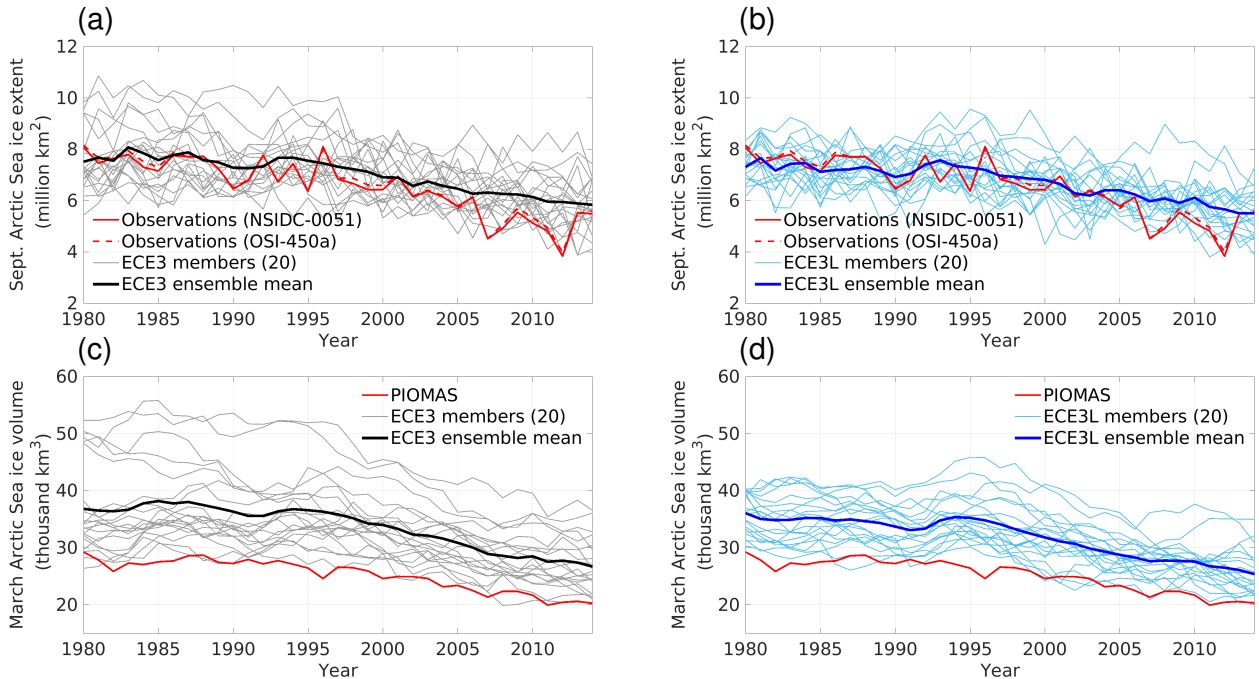

**Figure 8.** Arctic sea ice time evolution in the transient climate (1980–2014): (a) September extent (SIC > 15 %) ensemble of ECE3, (b) as in (a) but for ECE3L. Observations are from NSIDC with area pole-filling as well as OSI-450a. All are remapped to NSIDC-0051 grid. (c and d) as in (a and b) but for March volume (following the definition of the PIOMAS domain for areas thicker than 0.15 m).




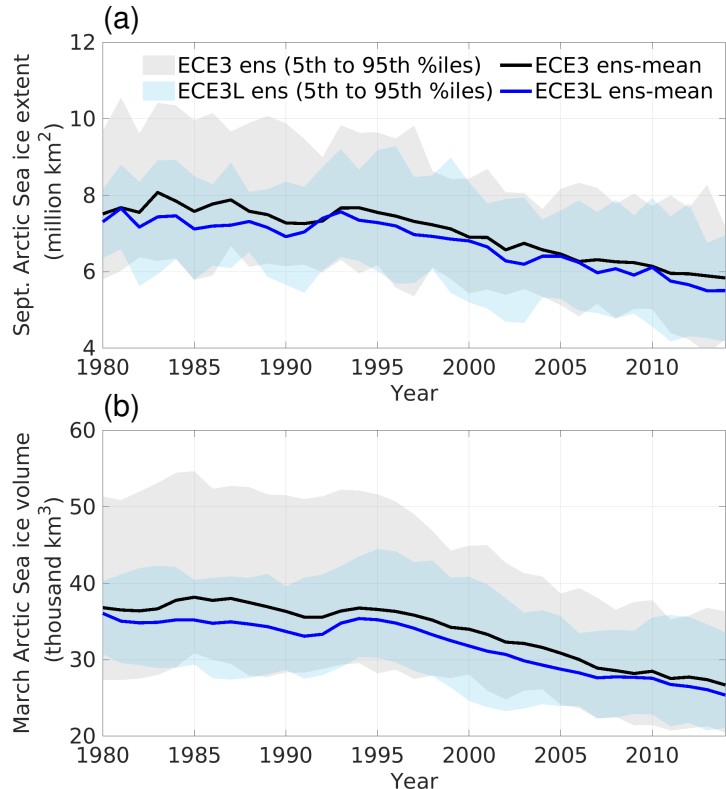

**Figure 9.** ECE3 and ECE3L Arctic sea ice (1980–2014): (a) September extent ensemble mean and 5th to 95th percentiles of 20 members, (b) March volume ensemble mean and 5th to 95th percentiles of 20 members.





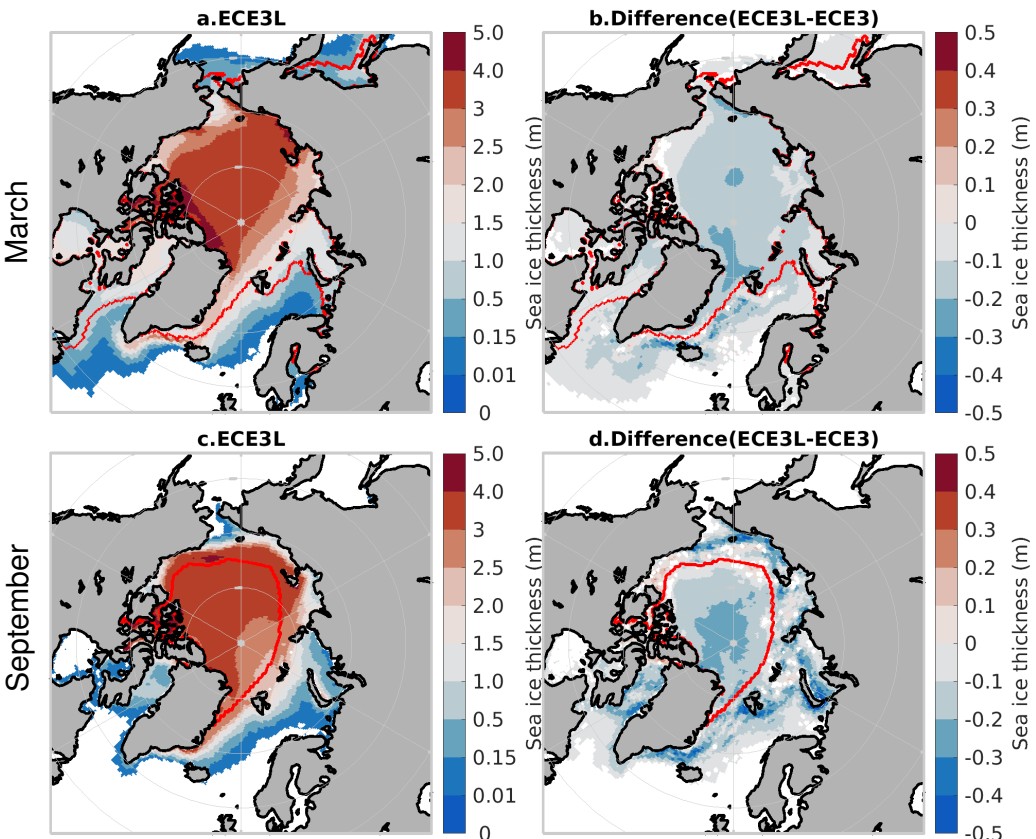

**Figure 10.** Ensemble mean Arctic maps (1980–2014): (a) ECE3L March sea ice thickness and (b) ECEL minus ECE3 difference. (c) and (d) as in (a) and (b), but for September. Note: nonlinear color scale is used in (a) and (b) to emphasize thin ice categories. Thickness under 0.01 m is not shown. Note that the PIOMAS domain is defined as SIT>0.15 m (see colorbar). The areas with SIC≥70% in ECE3L is compassed by red lines.





**Figure 11.** Integrated Ice Edge Error (IIEE, defined in section 2.3) maps of ECE3L (a) and ECE3 (b) vs. NSIDC-0051 for March sea ice climatology (1980-2014). Red and blue indicate whether the model's ensemble mean overestimates or underestimates the ice edge prescribed by NSIDC-0051, respectively. (c) and (d) as in (a) and (b), but for September sea ice. Sea ice edge is defined by the 15%-sea ice concentration contour.

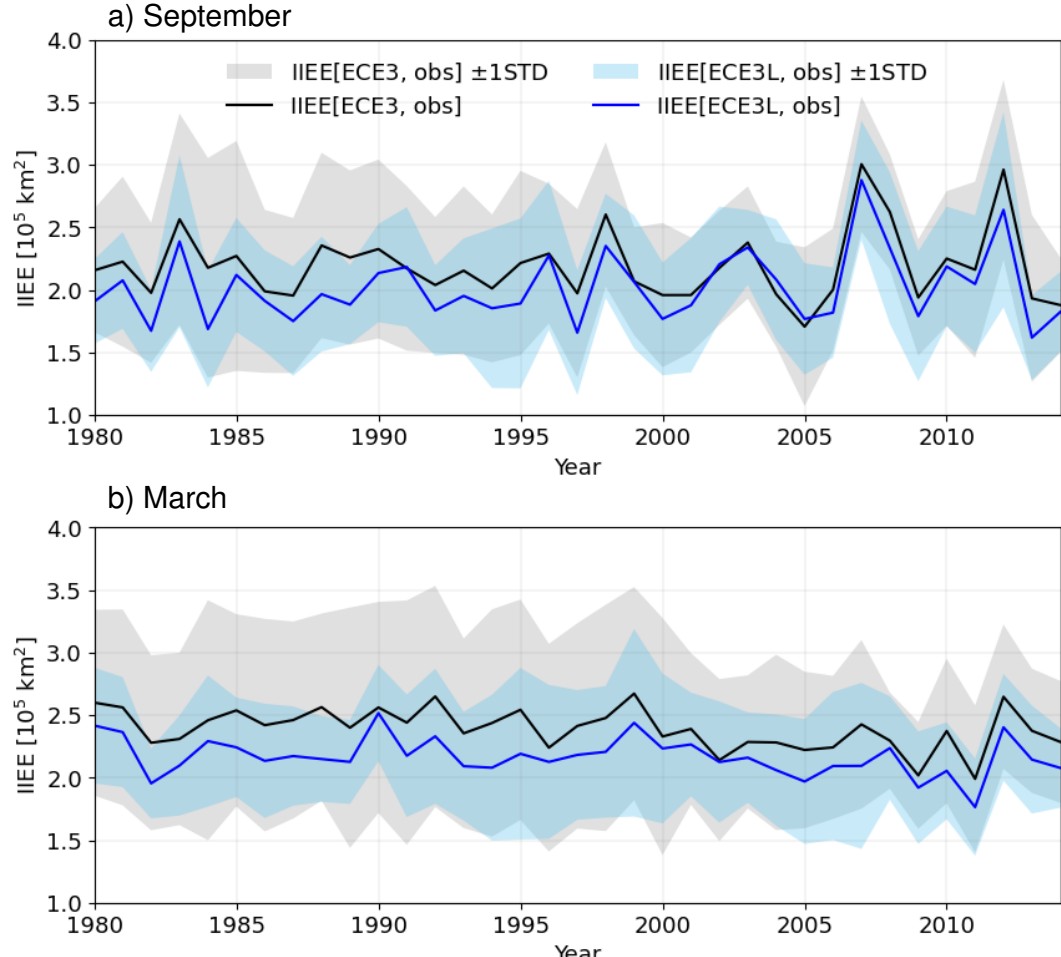

**Figure 12.** The time evolution of Integrated Ice Edge Error (IIEE) for September (a) and March (b) estimated for ECE3 (black) and ECE3L (blue) relative to NSIDC-0051 during the period 1980-2014: with ensemble means (thick line) and the model spread (the shaded area) indicated as one standard deviation from the mean across 20 members. Sea ice edge is defined by the 15%-sea ice concentration contour. The IIEE for each month is shown in Fig. S9.





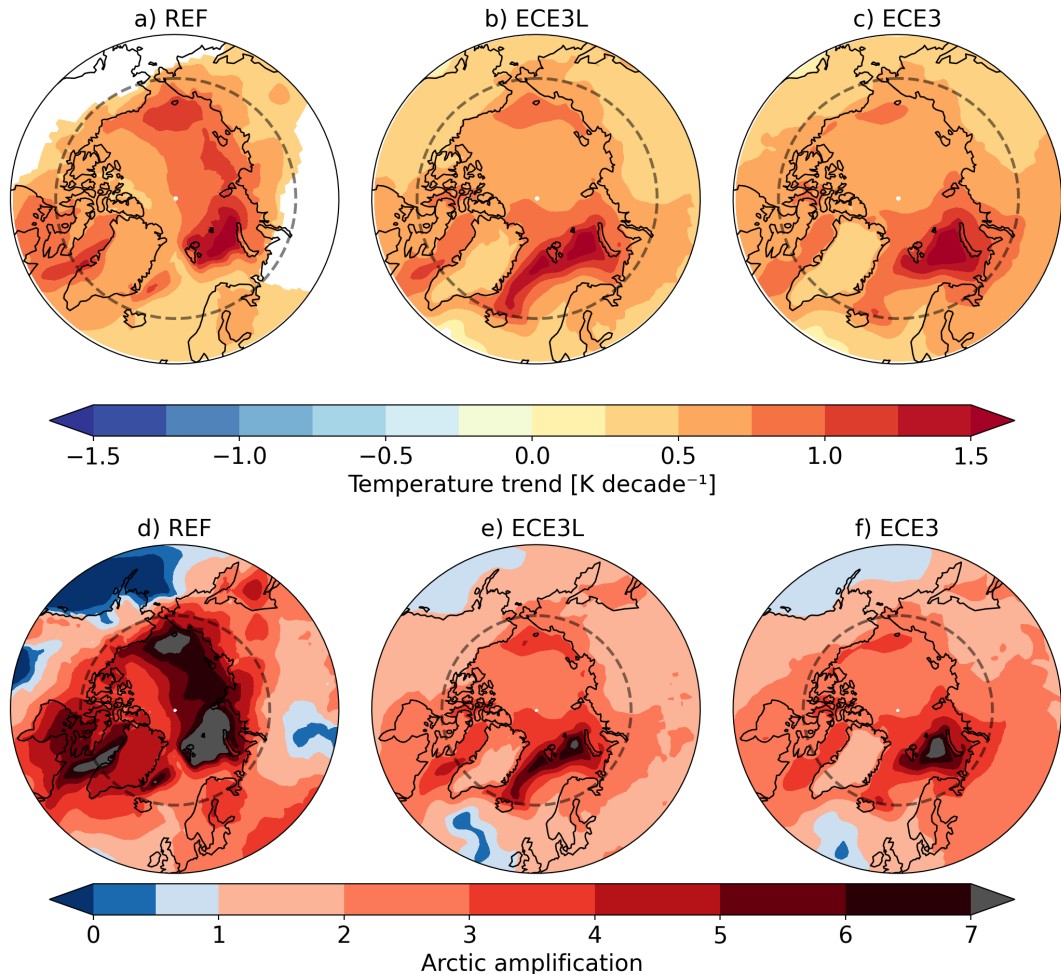

**Figure 13.** (a-c) Annual mean temperature trends for the transient climate (1980–2014), derived from the average of the observational datasets, ECE3L and ECE3. Areas without a statistically significant change are masked out. (d-f) as in (a-c), but for local amplification ratio, relative to the global mean temperature trend. The dashed line depicts the Arctic Circle (66.5°N latitude). Observations are from ERA5, BEST, JRA-55 and NCEP2.

  



## Appendix A: Empirical relations to modulate surface heat flux over sea ice

According to the method proposed by Davy and Gao (2019), the depth of convective boundary layer (in meters) is determined using Eq.A1, where the key parameter from the atmospheric model is the atmospheric stability at a height of around $300\,\mathrm{m}$ $\frac{dT}{dz}$ (in K m$^{-1}$). In the EC-Earth3 model configuration, temperature changes between model levels 86 and and 91 are used. These levels correspond to heights in the atmosphere approximately ranging from 300-500 m from the Earth's surface. The empirical relation in Eq.A1 is then adopted using constant values of $a_1$=230 and $a_2$=2100, so that the resulting $\lambda_{CBL}$ typically falls within the range of 1600 to 2500 m. It is important to note that the constants $a_1$ and $a_2$ have been adjusted to the temperature gradient between the specified model levels in the configuration of the EC-Earth3 model. These values were chosen to be consistent with the results from the original LES simulations of Esau (2007), and they will be different if different model levels or atmospheric models are employed.

$$\lambda_{CBL} = a_1 \frac{dT}{dz} + a_2. \tag{A1}$$

When the fraction of open water is less than 10% (i.e. SIC$\geq$90% ), it is featured as narrow leads. The modulating factor $A_{max}$ is empirically related to $\lambda_{CBL}$, where $a_3$=6.012e-8, $a_4$=-4.036e-4, and $a_5$=1.56, with a cap of [0.8 1.2].

$$A_{max} = a_3 \lambda_{CBL}^2 - a_4 \lambda_{CBL} + a_5. \tag{A2}$$

When the fraction of open water is larger than 30% (i.e. SIC$\leq$70%), the contribution from narrow leads is negligible and therefore the modulating factor is a constant of 1. Then for the open water fraction between 10-30% (i.e. 70%<SIC<90%), the effective factor $A_{lead}$ is linearly interpolated between 1 and $A_{max}$ as defined in Eq.A3.

$$A_{lead} = 1 + (A_{max} - 1) * (max(0, (SIC - 70))) / (90 - 70) \tag{A3}$$

Figure S1 shows examples of A$_{lead}$ for ExpCold and ExpWarm in the Arctic, respectively. This calculation is based on the lapse rate and sea ice concentration from previous atmosphere-only (AGCM) simulations in the Blue Action project with EC-Earth3 (Liang et al., 2020). The AGCM simulations were forced by historical forcing of CMIP6 and the surface boundary conditions from global daily 1/4 degree SSTs and sea ice concentrations (SICs) from the United Kingdom Met Office Hadley Centre Sea Ice and SST Version 2.2.0.0 (https://www.metoffice.gov.uk/hadobs/hadisst2/, Titchner and Rayner, 2014). The modulation effects exhibit remarkable seasonal variation, influenced by the background atmospheric stability, in the Arctic regions (see Fig. S1). During the winter months, there is an additional heat flux from the warm ocean through the leads to the atmosphere by means of the modulating factor above 1. Whereas during the summer months, the surface heat flux through the leads is reduced because the modulating factor is below 1. The areas where the lead parameterisation takes effect (SIC>70%) and are covered by ice thicker than 2 m have experienced substantial reductions from ExpCold to ExpWarm, particularly in the summer (seen in Figs. S1b,d). This suggests that the sea ice state, along with the presence of leads, has undergone significant changes during this period. These seasonal and interannual variations in the impact of leads on heat flux through the sea ice play a crucial role in the dynamics of the Arctic regions and their response to changing climate conditions.