# Peer review of "Modulating surface heat flux through sea ice leads improves Arctic sea ice simulation in the coupled EC-Earth3"

_EGUsphere, 2024_

## Author Comment (AC2)

**Reviewer #2:**

Citation: https://doi.org/10.5194/egusphere-2024-1865-RC2

**General:**

The manuscript aims at improving the Arctic sea ice state by introducing a modulated factor for sensible heat flux (based on Davy and Gao, 2019) depending on ice lead characteristics and the atmospheric boundary stability (enhancing in winter stable conditions and reducing it in summer unstable conditions). The scheme is shown to be effective during cold climate, although its impact seems to be less in normal or warm climate. The paper is well written. Unfortunately, some important details are missing and prevent a full acceptance of the manuscript at this stage. The heat exchange is not clearly established in view of the modulating factor and the cold and warm experiments are not clearly defined.

**Major comments:**

1.L87: "We emphasise that in ECE3L, the ocean does not supply additional heat to warm the atmosphere" (z for emphasize?) I am still scratching my head on this. However, no substantial addition is made to clarify the statement. This manuscript clearly states that the factor is applied to surface sensible heat flux: (L77-78) "we introduced a factor Alead to the surface sensible heat flux (SSHF) within the coupled ECE3 framework, to better represent the heat exchange through leads in sea ice ». Does it mean that the heat exchange between air and ocean-ice is non conservative? According to Davy and Gao (2019), only the heat flux over open water should be amplified but I was not able to ascertain the exact method used in the manuscript. Also, why stopping at sensible heat? Latent heat should be also pretty high over ice leads. Some results from Davy and Gao (2019) could be added in appendix since theirs is a project report to their funding agency (i.e., not clear whether it was peer-reviewed). Moreover, the details could be moved to the main text rather than the appendix since it is the core of the paper.

We apologize for any confusion caused by our statement. When we emphasized that in ECE3L, "the ocean does not supply additional heat to warm the atmosphere," our intent was to clarify that the heat extracted from the ocean per square meter, under the same air-sea temperature difference, remains consistent according to the bulk formula, regardless of whether it's in an open ocean grid cell or a grid cell with a small fraction of open water surrounded by sea ice. This implies that the ocean is not more prone to freezing, as the same amount of heat is transferred to the atmosphere. The introduced modulating factor, Alead, primarily affects the efficiency of heat transfer received by the atmosphere through the "amplified/damped" surface sensible heat flux (SSHF) over leads surrounded by sea ice. Consequently, this sub-grid process is non-conservative. (As reply to Reviewer #1, minor comment 9)

Davy and Gao (2019) explained that the heat flux over sea ice leads (characterized as small open water fraction within the grid cell) should be amplified. Specifically, we implemented this as at SIC >70% in a grid cell, the SSHF calculated over the fraction of open water will be amplified by the modulation factor.

We acknowledge the importance of latent heat flux over ice leads. We will address this limitation in the discussion section of our revised manuscript.

"*This is a sensitivity study, with a focus on the sensible heat flux. Since we didn't have information on how latent heat fluxes change in response to variations in lead width from the original LES simulations of Esau (2007), we chose not to make an assumption about how they would respond, but rather to quantify the sensitivity to changes in the sensible heat flux based on these LES simulations.* "

It is correct that Davy and Gao (2019) is a project report and non-peer-reviewed, while it provides a valuable foundation for our study. To address this, we will include key methodological details from Davy and Gao (2019) in the main text. We will incorporate relevant figures and findings from Davy and Gao (2019), particularly the results from the LES study as to how the fluxes from leads depend upon lead width under different atmospheric stabilities and how that was combined into a single amplification factor based on a lead width frequency distribution [Figures 1 and 4 from Davy and Gao (2019)], into the appendix to provide further context and support for our approach. We hope this will help clarify the methodology and ensure that the heat exchange process is more clearly understood.

2.L99 states that "The [cold and warm climate] simulations used constant forcing with a repeating seasonal cycle corresponding to the respective climate states ». However, I am still scratching my head on how you do this. In a coupled model, you have only variations is solar radiation at the top of the atmosphere and aerosol forcing. Nudging perhaps of one of the components? Please give more detail.

Here for the "constant forcing" of a specific year we mean to use the external forcing from CMIP6 historical forcing from the given year (including solar radiation, GHGs concentrations, aerosols and land use, ect) in the 50-year coupled simulation. The ocean and atmospheric variables still freely evolve in the simulation without being constrained.

3.Methodology: Some figures show the ensemble envelop (5-95% percentile), which is interesting since it gives us the statistical significance of the results. However, except for ice volume, they seem to be hardly significant (i.e., an overlap is visible), which needs to be stated in the text.

We acknowledge the importance of carefully interpreting the results. To address this, we have performed a paired *t*-test to assess statistical differences and will include a statement in the text to clarify this point. Specifically, we conducted the paired samples *t*-test in Python with Scipy library containing the ttest_rel() function. The resulting *p*-values have been provided for Fig.2, Fig.7 and Fig.9, in response to minor comments 5, 6 and 7, respectively. Additionally, we have assessed the significance for Fig. 12 as well.

4.L194-195 states that "The findings suggest that during Arctic winters, the decrease in sea ice concentration is mainly due to heat advected by the ocean from the south, rather than air-sea heat exchange through sea ice leads ». I think I see the same ice reduction in S6 for

concentration below 70% but I am not sure I understand the statement any better. Since the modulation factor is one over these regions, it must a non-local effect as mentioned by the authors. However, since we are dealing with a coupled system, it could be the ocean or the atmosphere. I am less inclined to think it is the ocean, as the authors do, since you would need to explain a change in the Arctic ability to pump more Atlantic waters northward, than a simple advection/diffusion of the warmer atmospheric boundary layer southward. So, please elaborate.

We acknowledge the reviewer's point. Combining our response to Reviewer #1, comment 13, we will revise this section.

*"The findings suggest that during Arctic winters, the overall thinning of sea ice, particularly at the ice margins, is a key driver of sea ice concentration reduction. In these marginal zones, where the ice is already very thin, even small reductions in thickness can lead to significant decreases in ice extent. In contrast, in the Central Arctic, sea ice concentration can remain close to 100% during winter, despite thickness reductions of one meter or more. This thinning at the margins coincides with a significant rise in surface air temperature by approximately 2 degree (Fig. 5), indicating a warmer atmospheric boundary layer extending southwards."*

**Minor comments:**

1.L56: Essential Climate Variables : why the capitalization here?
We have removed the capitalization of "Essential Climate Variables".

2.L248: what was the concentration threshold value used to define the ice edge? Please add.
As requested, we added *"with a 15% threshold for the sea ice edge (Goessling et al., 2016)"* to L248, in addition to its mention in L141, Section 2.3 Validation Data and Metrics.

3.L1 of Fig1 caption: please remove "compared"
It's removed.

4.L2 of Fig1 caption: "based on" I think you meant "relative to", i.e., the bars and maps are the difference between ECE3L and ECE3 base runs (?) for the two climate experiments.
Thank you for your comment. The term "based on" has been changed to "in" to clarify that only the output of the ECE3 baseline simulations is used for the analysis in the figure, rather than indicating a comparison of relative differences between ECE3L and ECE3 runs.

The figure compares surface temperature conditions between cold and warm climate states in the ECE3 model. It illustrates both the seasonal variations (Fig. 1a) and the spatial extent of sea ice cover (with SIC>70%, Figs. 1b and 1c). When (Fig. 1a) and where (Fig. 1b,c) the temperature difference is positive, an amplification factor (Alead >1) should be implemented to the ECE3L model, reflecting the reduced effect of the modulation factor in the warmer climate state.

5.Fig2: the volume does show statistical differences but not the area (at least not clearly). To be mentioned in the text.
We performed paired t-tests for Fig. 2a, b, c, and d to evaluate the statistical significance of the differences. The results will be included in the text: *"The mean differences are statistically significant, except for Fig. 2c."*

In Fig. 2a, ExpCold-SIA differences are significant (p = 2.25 × 10⁻⁹);
in Fig. 2b, ExpCold-SIV differences are significant (p = 1.60 × 10⁻¹⁶).
In Fig. 2c, ExpWarm-SIA differences are not significant (p = 0.083), while
in Fig. 2d, ExpWarm-SIV differences are significant (p = 0.010).

6.Fig.7: can the authors add the spread? I am worried that the significance is less than visually shown.

We have added the spread to Fig. 7. The ensemble means show significant differences for both Arctic sea ice area (p = 1.47 × 10⁻¹⁰) and sea ice volume climatologies (p = 3.35 × 10⁻¹⁷).

[Figure]

[Figure]

**Figure 7.** Comparison of seasonal cycle in the transient climate (1980-2014) between ECE3 (black) and ECE3L (blue) for (a) Arctic sea ice area and (b) Arctic sea ice volume. Thick lines represent the ensemble means, while the shaded areas indicate the spread between the 5th and 95th percentiles across 20 ensemble members, grey for ECE3 and light blue for ECE3L. Observations for the sea ice area include NSIDC and OSI-450a datasets, both remapped to the NSIDC-0051 grid. Sea ice volume is based on PIOMAS domain criteria (thickness > 0.15 m).

7.Fig.9: The plots do not show a statistically different mean (the two envelops overlap).

We assessed the statistical differences. The results will be included in the revised version. "*In Fig. 9a, the ensemble means for September Arctic sea ice extent are significantly different (p = 2.61 × 10⁻¹¹), and in Fig. 9b, the means for March sea ice volume also differ significantly (p = 4.99 × 10⁻¹⁷). These differences are primarily driven by external forcing. After removing linear trends, the residuals for both September SIE and March SIV are no longer significantly different (p > 0.05).*"

Additionally, in Figure 12, the time series of Integrated Ice Edge Error (IIEE) shows significant differences for both March (p-value = 1.78e-15, p < 0.05) and September (p-value = 2.27e-08, p < 0.05). No significant trends were detected in the time series.

8.L1 of Fig.10: ECEL should be ECE3

We corrected it as noted.

9.Fig.11: the solid circle appearing in the key of the map is misleading. Its interior should be lightly colored as the area covered.

We adjusted the figure legend to Fig.11, as suggested.

[Figure]

**Figure 11.** Integrated Ice Edge Error (IIEE, defined in section 2.3) maps of ECE3L (a) and ECE3 (b) vs. NSIDC-0051 for March sea ice climatology (1980-2014). Red and blue indicate whether the model's ensemble mean overestimates or underestimates the ice edge prescribed by NSIDC-0051, respectively. (c) and (d) as in (a) and (b), but for September sea ice. Sea ice edge is defined by the 15%-sea ice concentration contour.

10. Fig.13 and L291-295: Despite the authors' vigorous statement that ECE3L Arctic temperature trend is better than ECE3, both modelled trends are still quite far from the observed one. So, we are still missing out on the amplification factor in climate simulations. I realize that it is touched upon in the previous paragraph L287-290 discussing S10 but I must say that the S10a was not showing clearly this underestimation (but it is clearer in Fig.13), unless we are talking about something subtle hidden behind the spread of the ensemble? Why would there be a compensation between a global overestimation and the Arctic underestimation? Can you elaborate on this please, and possible in view of what the authors intent with the supplement (see below)?

Using the time series from Fig. S10a, we applied CDF to both visually and statistically assess the differences in variability and central tendency between the two ensembles (ECE3 and ECE3L) for the Arctic region (north of 66.5°N). As shown in Fig. 13, the Greenland-Iceland- Norwegian Seas (GIN: 40°W-15°E, 66.5°N-82°N) and the Barents and Kara Seas (BAKA: 15°E-100°E, 70°N-82°N) exhibit relatively remarkable changes in temperature trends and amplification ratio, compared to the rest of the Arctic domain. We applied CDF to the two key sub-regions.

[Figure]

**Figure S10.** Cumulative Distribution Function (CDF) plots for surface air temperature trend from ECE3 and ECE3L ensembles: (a) in the Arctic region (north of 66.5°N), (b) the Greenland-Iceland-Norwegian Seas (GIN: 40°W-15°E, 66.5°N-82°N), and (c) the Barents and Kara Seas (BAKA: 15°E-100°E, 70°N-82°N). Observations are from ERA5 (Hersbach et al., 2020), Berkeley Earth (BEST, Rohde and Hausfather, 2020), JRA-55 (Kobayashi et al., 2015) and NCEP2 (Kanamitsu et al., 2002).

In the Arctic, ECE3L simulates trends that are not only closer to the observed values but also with less variability compared to ECE3. This indicates that ECE3L more accurately captures the overall warming in the Arctic. In GIN, ECE3L exhibits a narrower distribution of temperature trends, demonstrating a closer alignment with the reference observational datasets. ECE3, in contrast, shows greater variability and tends to predict slightly higher warming trends than the observed references. In BAKA, ECE3L performs better than ECE3 by closely aligning with the observed warming trends. While some ECE3L estimates exceed 1.0 K/decade, it still provides a more accurate and consistent representation of temperature trends in this region compared to ECE3. Overall, these results suggest that ECE3L improves the representation of Arctic warming trends, particularly in terms of reduced variability and better alignment with observed data across key Arctic regions.

Because Fig.S10b shows indistinguishable differences of global mean T2m anomaly between ECE3 and ECE3L, we decided to remove the original Fig.S10a,b and replace them with the CDF plots.

**Appendix:**

11.L480: units are missing from a1 and a2.

We added units: a1 in units: $m^2K^{-1}$ and a2 in units m.

12.L481: is it not too large for winter Arctic Atmospheric Boundary Layer? Maybe add the expected and reasonable values for comparison.

We will add an explanation for the two extreme conditions in the appendix. *"This range of λCBL was derived from the original LES simulations which were tested on extreme cases of strongly stable stratification (30.7 K/km) to weak stratification (9.7 K/km). We therefore concluded that this was not an unreasonable limit to the resultant values for λCBL as we did not find more strongly stable stratifications than this in the lowest model level."*

13.Supplement figures: What is the goal of them: do the authors intent to keep them there (please revise the captions then with attention), or ultimately discard them?

Thank you for the suggestion. We will revise the figure captions to highlight the purpose of supporting materials and withdraw unimportant figures.

Fig. S1: We will add *"The figure illustrates the modulation of turbulent heat fluxes over sea ice, showing remarkable seasonal variation and such effect in different magnitudes between ExpCold and ExpWarm scenarios, particularly in regions with sea ice thicker than 1 m."*

Fig. S2-S3: As response to reviewer #1, major comment 2, we merged Fig.S2 and S3 to one and added the transient (r5) simulation (historical + SSP2-4.5, red) experiments from 1965-2065, to illustrate the effect of initial sea ice states and external forcing. We will add *"Figure S2. Yearly mean Arctic sea ice area (a), Arctic sea ice volume (b) and global 2-meter air temperature (c) from 1965 to 2065, comparing the ECE3 (black), ECE3L (blue), and transient (r5) simulation (historical + SSP2-4.5, red) experiments. Results from the 1985-forcing experiments are shown for the period 1985–2035, while results from the 2015-forcing experiments cover the period 2015–2065.*

*The changes in sea ice area and volume from ECE3 to ECE3L shows notable differences between the ExpCold and ExpWarm setups, revealing the sensitivity of sea ice evolution to initial seat ice conditions and external forcing, with relatively more sea ice reduction in ExpCold. The paired simulations are stable over the 50 year period in both forcing experiments, with no significant warming trends attributable to initialization artifacts. This stability is reflected in the absence of noticeable temperature drift in both forcing periods. In contrast, the transient simulation (r5), driven by historical and SSP2-4.5 external forcings, shows a clear warming trend over time. The small fluctuations in ECE3 and ECE3L are consistent with internal variability, indicating that the model does not exhibit any pronounced initialization-induced warming."*

Figure S4: We will add *"In summer the regions experiencing the greatest SIT reduction remain consistent with those identified in the winter, as shown in Fig. 3: from the eastern Arctic in the thicker ice regions in ExpCold shifting to the western Arctic in the thinner ice regions in ExpWarm."*

Figure S5: We will add *"In summer the regions experiencing the greatest SIC reduction remain consistent with those identified in the winter, as shown in Fig. 4. These are primarily in the thinner ice regions, typically along the ice margins, with up to 30% reduction in ExpCold compared to less than 20% in ExpWarm."*

Figure S6: We will add *"The regions showing the greatest SIC reduction in March and September climatologies are similar to those observed in ExpCold (Figs. 4a,b and S5a,b), though the magnitude of the reduction is lower compared to ExpCold."*

Figure S7: We will remove this figure, as the comparison with OSI-450a observations leads to the same conclusion as Fig. 11.

Figure S8: We will remove this figure for the same reason as Figure S7.

Figure S9: We will remove this figure for the monthly comparison of IIEE time series, as ECE3L consistently shows lower errors in sea ice edge representation across all seasons. We think Fig.12 is sufficient to present the IIEE for September and March, highlighting the errors during the sea ice minimum and maximum conditions.

Figure S10: As reply to comment 10, we will replace the figure of annual mean evolution of Arctic and global T2m anomalies from ECE3 and ECE3L ensembles with the CDF plots for the Arctic, the GIN seas and the BARA seas.

*The CDF analysis visually and statistically captures differences in the variability and central tendencies between the two ensembles. In the Arctic (a), ECE3L shows a closer match to observed trends, reflecting reduced variability and a more accurate central tendency compared to ECE3. In the GIN and BAKA seas (b&c), showing relatively pronounced changes in Fig.13, the CDF analysis highlights regional differences in ensemble performance, with ECE3L again demonstrating better alignment with observations. These plots provide insight into how each ensemble simulates warming trends in both the Arctic as a whole and in key sub-regions, helping to understand spatial variations in model performance.*

**Reference:**
Davy, R. and Gao, Y.: Improved key process in representing Arctic warming (D3.5), https://doi.org/10.5281/zenodo.3559470, The Blue-Action project (the European Union's Horizon 2020) Research and Innovation Programme under Grant Agreement No.727852., 2019.
Goessling et al.: Predictability of the Arctic sea ice edge, Geophys. Res. Lett., 43,1642–1650, 2016.
Hersbach et al.: The ERA5 global reanalysis, Quart. J. Roy. Meteorol. Soc., 146, 1999–2049, 2020.
Kanamitsu et al.: NCEP-DOE AMIP-II Reanalysis (R-2), Bull. Amer. Meteor. Soc., 83, 1631–1644, 2002.
Kobayashi et al.: The JRA-55 reanalysis: General specifications and basic characteristics, J. Meteor. Soc. Japan, 93, 5–48, 2015.
Rohde and Hausfather: The Berkeley Earth land/ocean temperature record, Earth System Science Data, 12, 3469–3479, 2020.

---

## Author Response (AR1)

**Reviewer #1:**

Citation: https://doi.org/10.5194/egusphere-2024-1865-RC1

**General:**

The submitted article "Modulating surface heat flux through sea ice leads improves Arctic sea ice simulations in the coupled EC-Earth3" by T. Tian, R. Davy, L. Ponsoni, S. Yang, provides interesting results on the effect of surface heat flux through leads on Arctic climate representation in one global climate model. It highlights the importance of representing the effect of small-scale processes.

The study is in most parts well written. The figures are nice and easy to understand. However, some of the conclusions from this study should be better supported by additional analysis or better explanation.

I thus recommend to accept this submission after revision that considers a few major and a somewhat larger number of minor comments.

Thank you for your thorough review and insightful comments on our manuscript titled "Modulating surface heat flux through sea ice leads improves Arctic sea ice simulations in the coupled EC-Earth3." We appreciate the opportunity to clarify our findings and address your concerns.

We have revised the manuscript and included a **marked-up version**, where removed text is shown in red and new text in blue. For your convenience, we have also updated the line numbers and new figure numbers in our previous point-by-point reply to align with the **marked-up** version.

**Main comments**

1. The authors highlight that the modulation is not only affecting Arctic sea ice but also improving the sea ice representation in EC-Earth3. They set up three improvement goals in the introduction (reduced sea ice, better trend, and Arctic ice minimum in September instead of August). These goals are only partly met. Sea ice is slightly less extended and slightly thinner in the improved model version when averaging over 1980-2014. However, August is still the month with lowest sea ice extent and the trend is not improved. The heat flux modulations seem further to have little impact in a present day climate but is more pronounced in a colder climate, and would likely have rather little impact in a warming climate.

Further, the authors do not show that the modulation of the heat flux is improving the heat flux in high-ice covered areas or atmospheric stratification in EC-Earth. This should be done to the extent possible. Otherwise, any potential improvement of the sea ice could be due to a compensation of errors. We should have in mind that many other processes including large scale ocean and atmospheric circulation are strongly affecting sea ice.

As long as it is not well shown that the modulation of heat fluxes is really improving the related local processes, this study, which provides interesting and relevant results, should be seen as a sensitivity study to understand the impact of modulating the heat flux over leads, and not try to sell it as an improvement that solves long-standing biases in global models.

We appreciate the careful evaluation of our results. Regarding the Arctic warming trend over 1980–2014, we applied cumulative distribution functions (CDFs) to both visually and statistically assess differences between the two ensembles for the Arctic and key sub-regions in Fig.13d,e,f. These analyses suggest that ECE3L offers an improved representation of Arctic warming trends, particularly by reducing variability and showing better alignment with observed data. This is addressed in response to minor comment 6. We also showed improvement in representing the declining rate of sea ice volume in March in response to minor comment 8.

We acknowledge the lack of observational data on how sensible and latent heat fluxes respond to variations in lead width in Arctic regions. Given this limitation, we avoided making assumptions about these responses. Instead, we quantified the sensitivity of the sensible heat flux based on the LES simulations conducted by Esau (2007). We revised the title accordingly "*Impact of modulating surface heat flux through sea ice leads on Arctic sea ice in EC-Earth3 in different climates*". In both the abstract (L1) and discussion (L418), we emphasize that this is a sensitivity study, highlighting the potential contribution of sea ice lead parameterization in addressing long-standing biases.

Additionally, we compared the winter mean turbulent heat flux between the ensemble means and ERA5 in Figure 1. The ECE3 ensemble significantly underestimates the upwards heat flux at the sea ice margins, whereas the ECE3L ensemble slightly enhances the heat flux, aligning with the area with sea ice reduction in March, as shown in Fig. S6b.

[Figure]

(a) **ECE3 - ERA5**     (b) **ECE3L - ECE3**

Upwards Turbulent HF (W m$^{-2}$)

-25  -20  -15  -10  -5   0   5   10   15   20   25

**Figure 1.** The mean difference in upwards turbulent heat flux (the sum of surface sensible heat flux and surface latent heat flux) during winter (January to March) between **(a)** the ECE3 historical ensemble and ERA5, and **(b)** between ECE3L and ECE3, over the period 1980-2014. The ECE3 ensemble consists of 19 members, with missing realizations from r6, r9, r11, r13, r15, and r20. Stippling indicates areas that are not statistically significant. The blue line and pink line indicate the mean sea ice concentration above 70% and below 15%, respectively.

2. The cold and warm climate simulations are very short (section 2.2). Although Figures S2 / S3 indicate that the 30-year period chosen for analysis seem to be rather stable for sea ice area, volume and global air temperature, a 20-year spin-up is very short. I would expect, and earlier studies showed this, that climate warms (depending on the model by maybe around 0.2 - 0.6 degree C) after initializing from a transient run and repeating the forcing from that year before stabilizing. EC-Earth3 seems not to show such a warming after initialization from the transient run, or the warming is very small and short in time, or it is not visible due to internal variability and the shortness of the simulations. It should be checked which of these alternatives is true.

A 30-year period is also short for comparison between the cold and warm climate, particularly given that EC-Earth3 shows huge internal variability on centennial scales. Maybe, the somewhat surprisingly much larger effect of the modulating heat flux in ECE3L in the cold climate compared to the transient run can partly be explained by internal variability. I suggest to make the cold and warm simulations in total at least 100-year long to get more robust results.

We acknowledge the possibility of large internal variability in the global mean temperature (GMT) by around 0.2 to 0.6°C (Doescher et al., 2022), therefore, it is important to analyze the development of GMT over time.

To address the concern whether our experiment design shows a warming after initialization from the transient run, or can overcome the model internal variability, we modified Figure S3, by comparing the yearly mean GMT between the ECE3, ECE3L, and the transient simulation (r5, which provides the initial conditions for the 1985 and 2015 forcing experiments). We have now combined the original Figs. S2 and S3 into a single figure (referred to as Fig. S2, see below). In L243 & L252.

Specifically, we calculated the difference of GMT between the last 30-year mean (shown in original Fig.S3) and the first 30-year mean for each simulation, respectively. As shown in the new Fig. S2c, the

- For the 1985-forcing run, the difference is 0.06°C for ECE3 (0.11°C for ECE3L) and the 50-year trend is 0.03 °C/decade for ECE3 (0.05 °C/decade for ECE3L).
- For the 2015-forcing run, the difference is -0.03°C for ECE3 (-0.03°C for ECE3L) and the 50-years trend is -0.00°C/decade for ECE3 (-0.01 °C/decade for ECE3L).

Based on this analysis, it seems unlikely that the climate would warm by 0.2-0.6°C within the next 50 years due to initialization effects. The paired simulations are stable over the 50 year period in both forcing experiments, with no significant warming trends attributable to initialization artifacts. This stability is reflected in the absence of noticeable temperature drift in both forcing periods. In contrast, the transient simulation (r5), driven by historical and SSP2-4.5 external forcings, shows a

clear warming trend over time. The fluctuations in ECE3 and ECE3L are consistent with internal variability, indicating that the model does not exhibit any pronounced initialization-induced warming.

Regarding internal variability, we observed the following over the 30-year and 50-year periods:

- 1985-forcing: the standard deviation is 0.1°C for ECE3 (0.1°C for ECE3L) over 30 years, and 0.1°C for ECE3 (0.1°C for ECE3L) over 50 years.
- 2015-forcing: the standard deviation is 0.1°C for ECE3 (0.1°C for ECE3L) over 30 years, and 0.1°C for ECE3 (0.1°C for ECE3L) over 50 years.
- Transient run: the standard deviation is 0.2°C for 1985 (25 members) and 0.2°C for 2015 (24 members) in ECE3.

As internal variability in ECE3 and ECE3L is of a similar magnitude in their respective cold/warm climate scenarios and remains lower than that in the transient climate in the corresponding years, there is no evidence of amplified internal variability in ECE3L due to the modulated heat flux. Therefore, the larger effect of sea ice reduction in the cold climate, compared to the transient run, is unlikely due to internal variability.

[Figure]

**Figure S2**: Yearly mean Arctic sea ice area **(a)**, Arctic sea ice volume **(b)** and global 2-meter air temperature **(c)** from 1965 to 2065, comparing the ECE3 (black), ECE3L (blue), and transient (r5) simulation (historical + SSP2-4.5, red) experiments. Results from the 1985-forcing experiments are

shown for the period 1985–2035, while results from the 2015-forcing experiments cover the period 2015–2065.

While we acknowledge that a longer simulation period (e.g., 100 years) would provide more robust results, it is unfortunately not feasible to extend the original simulations, as the supercomputer used for this study has been decommissioned. However, as suggested, we now present the full 50-year time series of sea ice area and volume in Figure S2, to ensure a more comprehensive view of the model behavior over time.

As response to Reviewer #2 comment 13: we will revise figure captions to highlight the purpose of supporting materials in Fig. S2 as follows:
"*The changes in sea ice area and volume from ECE3 to ECE3L shows notable differences between the ExpCold and ExpWarm setups, revealing the sensitivity of sea ice evolution to initial seat ice conditions and external forcing, with relatively more sea ice reduction in ExpCold. The paired simulations are stable over the 50 year period in both forcing experiments, with no significant warming trends attributable to initialization artifacts. This stability is reflected in the absence of noticeable temperature drift in both forcing periods. In contrast, the transient simulation (r5), driven by historical and SSP2-4.5 external forcings, shows a clear warming trend over time. The small fluctuations in ECE3 and ECE3L are consistent with internal variability, indicating that the model does not exhibit any pronounced initialization-induced warming.*"

3. While it is very nice that the ensemble of historical simulations is large with 20 members, the method to initialize the ensemble of historical simulations from only two historical members of EC-Earth3 in 1960 might lead to an underestimation of the spread in the ECE3L ensembles compared to the original ECE3 ensemble. The large spread across CMIP6 ECE3 members due to long-term large internal variability (AMOC) might not sufficiently be captured by this method based on 2 members only. It would be good to see where these 2 members that are used to create the ECE3L ensemble are placed in the cloud of the original 25 ECE3-members (e.g. AMOC, global mean temperature, Arctic ice volume).

Thank you for your insightful comment. We acknowledge that internal variability in a coupled climate system can indeed arise from multiple components, including the atmosphere and the Atlantic Meridional Overturning Circulation (AMOC).

In our study, the two members were initialized from states reflecting moderate deviations from the ensemble mean of 25 members — r5 from a slightly colder state and r8 from a slightly warmer state (Fig.2a, pointed by red arrows). This initial difference between r5 and r8 is reflected in the AMOC behavior, TAS and SIV evolution during the historical period from 1850 to 1980 (Fig.2b-d). For the year 1960 the differences between r5 and r8 (with solid dots) are 3.8 Sv, 0.3K and -12.8 thousand km$^3$, respectively. We include the original 20 ECE3-members (data publicly accessible) in Fig.2b,c,d for the TAS, SIV, and AMOC time series, respectively. The AMOC strength, as well as other parameters in the historical simulations, exhibits considerable inter-annual variability. Specifically,

for AMOC, the magnitude of internal variability in both r5 and r8 is similar to that of other ensemble members.

[Figure]

**Figure 2. (a)** Adapted from Figure 3 in Doescher et al. (2022). Time series of the global mean of annual near-surface temperature (TAS) over a 500-year-long EC-Earth3 piControl experiment. The thick blue line represents an 11-year running average of the annual mean. The time axis is arbitrary due to the constant forcing applied throughout the experiment. Red circles mark the initial states from which the members of the historical experiment are initialized. The realization IDs of the historical ensemble members (r1-r25) are displayed at the bottom, and r5,r8 are highlighted by red arrows. **(b,c,d)** Annual global mean TAS, Northern Hemisphere sea ice volume (SIV) in March and annual Atlantic Meridional Overturning Circulation (AMOC) in the historical ECE3 simulations (thin lines, 20 members available) at 26°N. The r5 (blue) and r8 (red) members are indicated by thick lines, and the values in 1960 are marked by circles, respectively.

For the generation of the ECE3L ensemble, we introduced small random perturbations (on the order of $10^{-5}$ K) in the 3D temperature field. Although these perturbations are minor, they are adequate to induce divergence among ensemble members after only a few days.

We compared the ensemble mean and the model spread for TAS (a), SIVin March (b) and AMOC (c) between ECE3L and the original ECE3 ensemble in Figure 3 and Table 1. We applied paired *t*-test to assess statistical differences. In Fig. 3c, although the difference in AMOC between r5 and r8 has converged since 1980, the ECE3L simulations maintain a similar magnitude of model spread (std=1.4 Sv) as ECE3 (std=1.5 Sv) averaged over the period from 1980 to 2014 (Table 1). The ensemble means

of AMOC from 1980-2014 show significant differences between ECE3 and ECE3L (p < 0.05), indicating that our method captures the internal variability of AMOC adequately.

For GMT and SIV, their ensemble means are also significantly different with p<0.05, while the detrended time series are not significantly different with p>0.05, indicating the dominant role of external forcing.

In light of your comment, we add clarification regarding the representation of AMOC internal variability in ECE3L in L314-319 and Fig. 3a&c to Fig.S5 (also in response to comment 14).

[Figure]

**Figure 3.** (a-c) as in Figure 1. b-d, but for the ECE3 and ECE3L members (thin lines) over time, with ensemble means (thick line) and model spread (shaded area) represented as one standard deviation from the ensemble mean across 20 members. Note r5 and r8 are indicated by pink and red lines.

**Table 1.** Ensemble mean difference (ECE3-ECE3L) and ensemble spreads averaged over time (from 1980-2014) for ECE3 and ECE3L.

| Parameters | Mean difference (ECE3-ECE3L) | Mean model spread (ECE3) | Mean model spread (ECE3L) |
|---|---|---|---|
| GMT (K) | -0.1 | 0.2 | 0.2 |
| SIV ($10^3$ km$^3$) | 1.8 | 6.0 | 3.9 |
| AMOC (Sv) | -0.3 | 1.5 | 1.4 |

**Minor comments:**

1. Title: Since the improvement (reduction of sea ice) seems to be limited to the earlier part of the historical period that is analysed here, and there is little evidence that modulating the heat flux improves present day sea ice, I suggest to change the title. Maybe something like: "Impact of modulating surface heat flux through sea ice leads on Arctic sea ice in EC-Earth3."

Please delete the "coupled" before EC-Earth3. Either write the "global coupled climate model EC-Earth3" or only "EC-Earth3". To my understanding EC-Earth3 is as default coupled.
We revised the title accordingly. The new title of the manuscript is "*Impact of modulating surface heat flux through sea ice leads on Arctic sea ice in EC-Earth3 in different climates*".

2. Line 4: without reading the entire article it is impossible to know, what "cold" and "warm" climate refers to. "cold" sounds like PI or even colder and "warm" like some time in the future. I recommend to say something like: …one pair using 1985-forcing (cold climate) and the other 2015-forcing (warm climate).
Thank you for your valuable feedback. We revise it in L6 and add it to L467, Table 1, and Figs. S1-S2.

3. L7: "two CMIP6 historical ensembles". Sounds like an ensemble of historical simulations from different CMIP6-models. Make clear that you performed an historical ensemble with ECE3L and compare it to an ECE3 ensemble.
Yes, we rephrased it accordingly in L10 & L284.

4. L8: It is unclear why ECE3L is in () . I guess what you mean is that both the ECE3 and ECE3L ensemble means closely resemble the mean states in the respective cold-ensemble.
We realize that our original phrasing may have been unclear, and we rephrase the sentence in the abstract to improve clarity. Here is the revised sentence in L11:
"*We found that the spatial changes in mean sea ice states between the ECE3 and ECE3L ensemble means in the transient climate closely resembled those observed in the 1985-forcing (cold-climate) experiment. However, the magnitude of reduction in the total sea ice area and volume achieved by ECE3L relative to ECE3 was nearly four times greater in the cold-climate than in the transient-climate experiment, suggesting the diminishing role of sea ice leads in a changing climate with decreasing occurrences of stable stratification in winter.*"

5. L10/11: It does not seem very logical that the mean climate states are similar in the cold climate and the transient run, but the effect of the modulated heat flux so much smaller in the transient run. If the mean sea ice state of the transient run is similar to the cold climate, the argument that diminishing ice leads to a smaller impact of modulating the fluxes in the transient run does not make sense. In fact, your figures seem to show that sea ice is even a bit thicker in the transient run than in the "cold" climate.
We acknowledge the reviewer's point. As shown in Figs. 3a,b and 10a,b in the manuscript, the mean sea ice thickness (SIT) in March for ECE3L and the difference against ECE3 are provided for both the

1985-forcing and transient runs. The smaller reduction in sea ice during the transient run leads to thicker ice remaining in ECE3L (Fig.10a) compared to the cold climate (Fig.3a), as seen here in Figs. 5a (transient run) versus Figs. 4a (cold climate). For reference, ECE3's mean SIT is shown on the right panels in Figs. 4 and 5. The SIT in March is slightly greater in the central Arctic for the cold climate (Fig. 4b) than the transient run (Fig. 5b), with little difference in summer (Figs. 4d and 5d). More thinner ice (<0.5m) observed in the ice margin in the transient run in both seasons likely results from interannual variability. We revised L13 and, to support this statement, improved the clarity in the results section 4.2 (L321–332), particularly by incorporating this explanation into L329.

[Figure]

**Figure 4.** 30-year mean sea ice thickness in March (a,b) and September (c,d) in the cold-climate experiments.

[Figure]

**Figure 5.** As Fig.4, but for 35-year mean in the transient-climate experiments.

6. L11/12: I disagree to the statement that the local amplification rate in ECE3L is significantly improved compared to ECE3. Maybe in parts of the Greenland and Labrador Seas but in contrast ECE3 is clearly better in the Barents Sea. Please explain clearer (in section 5.1) why you would judge that this is a significant improvement or delete this sentence.

We acknowledge the reviewer's point. To address your concerns, we applied CDF to analyze the time series from the original Fig. S10a. This approach enabled us to visually and statistically assess the differences in variability and central tendency between the ECE3 and ECE3L ensembles for the Arctic region (north of 66.5°N). As noted that the Greenland-Iceland- Norwegian Seas (GIN: 40°W-15°E, 66.5°N-82°N) and the Barents and Kara Seas (BAKA: 15°E-100°E, 70°N-82°N) exhibit relatively remarkable changes in temperature trends and amplification ratio compared to the rest of the Arctic domain Fig. 13, we applied CDF to these two key sub-regions.

[Figure]

**Figure 13**. Cumulative distribution function (CDF) plots for TAS trend from ECE3 and ECE3L ensembles: (d) the Greenland-Iceland- Norwegian Seas (GIN: 40°W-15°E, 66.5°N-82°N), (e) the Barents and Kara

Seas (BAKA: 15°E-100°E, 70°N-82°N) and (f) the Arctic Circle. The locations of GIN and BAKA are shown in Fig. 1c.

In the Arctic (Fig.13f), ECE3L aligns more closely with observed trends, particularly by reducing variability and providing a more accurate central tendency compared to ECE3. In the GIN and BAKA seas, the CDF analysis highlights regional differences in ensemble performance, with ECE3L again showing better alignment with observations (Figs.13d&e). While the ensemble mean of ECE3 seems slightly better than ECE3L for temperature trends and amplification ratios in the Barent Sea (the original Fig.13c,f), the variability within individual members makes ECE3L more reliable regionally in Fig.13e. (Please refer to our response to Reviewer #2, comment 10, for further details.)

In summary, we deleted the original Figs. 13e–f showing the local amplification ratio. Accordingly, we revised the corresponding sections in abstract, the methods and results discussion (L17, L216, L377-381). We added CDF plots to Figs. 13e–f, discussed them in L388, and emphasized the findings in the conclusion (L472–475). **The new results align with our previous findings** on the local amplification ratio but provide stronger evidence through enhanced visual and statistical assessments.

7. L27: "most climate CMIP6 models struggle to reproduce the rapid decline since the mid-2000s" – this suggests that most CMIP6 models underestimate the observed trend but is this really true? The CMIP6-model mean (e.g. Notz and Community) slightly underestimates the trend between 2000 and 2014 but if we would consider the observed ice area until 2023 and compare versus e.g. hist+ssp2-45, the CMIP6 model ensemble mean is well representing the trend. Differences among models are large (Keen et al. 2021) but is Keen et al. 2021 really stating that CMIP6 models generally are underestimating the trend since mid-2000s?

Further, observations show a rapid decline until 2012 but no further decline thereafter, thus it "rapid decline since the mid-2000s" is not entirely correct from todays (year 2024) view.
We appreciate the reviewer's feedback and have clarified the following points in response:
- Rapid decline period: The rapid decline primarily occurred between the early 2000s and 2012, with the rate of decline slowing afterward (Lee et al., 2023; Sumata et al., 2023).
- Model performance: While the CMIP6 multi-model mean captures the overall trend, individual models vary in their ability to reproduce the decline. Differences in model performance are mainly attributed to biases in sea ice volume and growth processes (Keen et al., 2021).
- Feedback mechanisms: Key feedback processes, such as the sea ice-albedo effect, are often underrepresented due to biases in sea ice modeling, which introduces uncertainties in future projections (Wunderling et al., 2020).

Please modify this statement.
*"In addition, CMIP6 models exhibit significant inter-model variability in simulating Arctic sea ice decline, particularly during the accelerated loss that began around the early 2000s. This includes the*

*nonlinear shift from thicker, deformed ice to a thinner, more uniform regime after 2007 (Sumata et al., 2023). While the multi-model mean captures the observed sea ice trend until 2012, discrepancies arise in models' ability to simulate the timing and magnitude of the decline, with some underestimating or overestimating trends due to biases in sea ice volume and growth processes (Keen et al., 2021; Lee et al., 2023). These discrepancies challenge the reliability of future climate projections, particularly regarding sea ice loss (Wunderling et al., 2020). Consequently, critical feedback mechanisms, such as the ice-albedo effect that amplifies Arctic warming, cascade uncertainty, limiting models' ability to project reliable future sea ice evolution and its broader impact on global climate systems."* See L34.

8. L 52/53: This is an important question that is phrased here but I do not find any answer to it in conclusion or abstract of this article. To me it seems that your conclusion would be "no". The cold climate shows less ice/ is warmer with heat flux modulation but the warm climate shows only little change, thus the delta sea ice between warm and cold states is actually smaller in the ECE3L than the ECE3 runs. Surprisingly, this is not really reflected in the amplification rate or the trends.

Please take up the question again later, e.g. in the conclusions.
Thank you for your valuable feedback. We explicitly addressed this in the abstract (L15-19), the results (L311–314), and the conclusion (L469-475), based on the following analysis.

We have raised two questions: Q1. Whether the amplified heat flux through sea ice leads during winter can accelerate the transition from a colder state with thicker ice to a warmer state with thinner ice. Q2. Whether this amplified heat flux becomes less effective in a warming Arctic.

We hypothesized that higher SIV acts as a buffer, delaying the response to warming. This is because additional energy is required to thin thicker ice. By removing the buffer, we expected the model to better represent the thinning rate in the warming Arctic. To explore the transition from a cold-thick state to warm-thin state, we calculated the declining rate in SIV for two periods, 1980-1999 and 1995-2014 (Fig.6a). We also applied CDF to analyze the trends from 1980 to 2014 (Fig.6b)

[Figure]

**Figure 6. (a)** Time series of March SIV ($10^3$ km³) for PIOMAS (red), ECE3 (black), and ECE3L (blue) from 1980 to 2014. Linear trends are indicated for two periods: 1980-1999 and 1995-2014. The respective trends (in $10^3$ km³/decade) are:
    PIOMAS: -1.0 for 1980-1999 and -3.5 for 1995-2014;

ECE3: -1.0 for 1980-1999 and -5.5 for 1995-2014;

ECE3L: -0.9 for 1980-1999 and -5.0 for 1995-2014.

This panel shows that both ECE3 and ECE3L simulate stronger SIV reductions compared to PIOMAS during the latter period.

**(b)** CDF of the March SIV trend ($10^3$ km³/decade) for the ECE3 (black) and ECE3L (blue) ensembles from 1980 to 2014. The red vertical line represents the observed trend from PIOMAS. The CDF shows over 60% of the members in both ECE3 and ECE3L overestimate the SIV decline (left of the reference line), with trends in ECE3L generally closer to the observed trend and exhibiting reduced variability.

In summary, the amplified heat flux through sea ice leads in ECE3L does not accelerate the decline in SIV, but rather mitigates the overestimated decline rate observed in ECE3, which is driven by a more pronounced positive bias in SIV in colder states. ECE3L ensemble exhibits reduced variability and shows potential for improving the model's sensitivity to Arctic amplification in certain regions. However, long-term trends in SIV (Fig. 6b) and TAS (Fig. 13d-f) in the Arctic remain primarily driven by external forcing (as discussed in the response to major comment 3). The role of sea ice leads will diminish under future warming scenarios but remains crucial in colder climates, where ice loss and surface warming are more pronounced.

9. L87: "does not supply additional heat to warm the atmosphere". Please explain:

We revised it in L141. "*In ECE3L, the heat extracted from the ocean per square meter remains consistent for a given air-sea temperature difference, whether in open ocean or in grid cells with small open water fractions surrounded by sea ice, as determined by the bulk formula. This means the ocean is not more prone to freezing, as the same amount of heat loss to the atmosphere. The modulating factor, Alead , primarily adjusts the efficiency of heat transfer via "amplified/damped" SSHF over leads in sea ice, potentially making this sub-grid process non-conservative.*" (As reply to Reviewer #2, major comment 1)

Do you mean the amplification of heat flux in winter is compensated by below-1-values in summer, and in the annual mean there is no additional heat flux to the atmosphere? And why should not the heat flux into the atmosphere be increased in a coupled model if it would be more realistic?

Not exactly, it is that there is usually above-1 values in winter when you more often find stable stratification than there is in summer, but there is nothing prescribing that the net effect on heat fluxes in the annual mean should be 0; this depends upon the climatology of the atmospheric stratification in the model.

10. L 110-115: "These simulations were a subset of a 25-member ensemble." How many EC-Earth3 members are you using: 25 or 20 or what do you mean with "subset"?

We used 20 members, which are a subset of the original 25-member ensemble. We improved the clarity in L169-172.

11. L118: How do you perturbed the atmospheric initial conditions?

The 3D temperature field was perturbed with random differences (to the order of $10^{-5}$ K), which are much smaller than observations yet sufficient to generate enough differences in the different members after a few days. We improved the clarity in L179 and the caption of Fig. S5.

12. L177: "..in EXPWarm during the 50-year cycle (as revealed in the full time series in Fig S2)" . Figure S2 does not show the full 50-year time series but only year 21-50, please correct. As stated before, 100-year runs would provide more robust results, and please show really the entire time series as it would be very interesting to see how quickly the sea ice changes evolve in ECE3L.
We now present the full 50-year time series of sea ice area, volume and GMT in Figure S2. Please refer to our response to major comment 2, for further details (Now in L243).

13. L194/195: Do you want to indicate that ocean heat transport from the south into the Arctic increases in ECE3L, and that the increased heat is decreasing more sea ice? In case this is the hypothesis, please show ocean heat transports into the Arctic. However, I believe that it is more likely that sea ice is generally getting thinner. At the ice margins, ice is getting so thin, that it drastically drops while in the Central Arctic sea ice concentration will still be close to 100% in winter even if ice thickness is reduced by 1 m or more.
Your latter interpretation is correct.  Combining our response to Reviewer #2, comment 4, we revised this section in L262-268:
*"The findings suggest that during Arctic winters, overall sea ice thinning, particularly at the ice margins, is a key driver of sea ice concentration reduction. In these dynamic marginal zones, where the ice is often thin and fractured, even small reductions in thickness can lead to substantial decreases in ice extent. In contrast, the concentration in the central Arctic's pack ice remains close to 100% during winter, even with thickness reductions of one meter or more. This thinning at the margins coincides with a significant rise in surface air temperature by approximately 2 degree (Fig. 5), indicating a warmer atmospheric boundary layer extending southwards."*

14. L 230: Model spread is smaller in ECE3L than ECE3 -  this might be an artifact of the initialization methodology; see main comment 3.
As reply to major comment 3, we showed the ensemble mean and the spread of TAS and AMOC in ECE3L are comparable to those in the original ECE3 ensemble (Fig.3 and Table 1). We add clarification that the reduced model spread is due to the reduced positive bias in the colder conditions, unlike an artifact of the initialization methods (L314-319), as well as Fig. 3c to Fig.S5.

15. L235: "can refine estimates of the declining sea ice trend". Yes, but it would reduce the trend since the effect of heat flux modulation is larger in a colder climate. Is not that in contrast to one of your goals to make the trend in CMIP6-models or specifically in EC-Earth3 more realistic (= larger) ? (Although as stated before I disagree that CMIP6 models generally underestimate the observed sea ice trends).

As reply to minor comment 8, it reduces (rather than accelerates) the overestimated trend for ECE3. We've also clarified the CMIP6 models's performance in representing the declining rate, as response to minor comment 7. In abstract  L15-19, results L311–314, and the conclusion (L469-475).

16. L239-245: Linked to minor comment 5: Do you have any explanation why the impact of modulating heat flux in the transient period is so much smaller than in the cold climate, although the cold climate has even slightly thinner ice than the transient run?
We addressed this to minor comment 5 (In section 4.2 L321–332).

17. L270: "has significantly improved its accuracy …". But the improvement seems to be time-dependent and mainly for a (past) colder climate. If I follow your argumentation, then you would not expect large impact in a warmer future climate as well or would you? Any improvement of models is good but how relevant is this improvement to "improve predictions of future Arctic conditions" (Line 275) then?
We acknowledge the reviewer's point and agree that while the improvement is significant, the parameterization has limitations in warmer states, which may limit its impact on improving predictions of future Arctic conditions. As a result, we removed the phrase "*to improve predictions of future Arctic conditions*" from L367 and revised the context  accordingly in L358-363.

18. L291: "The temperature trend maps demonstrate that ECE3L outperforms ECE3": Linked to comment 6, I disagree: both ECE3L and ECE3 show trend patterns that are quite different from the reference data sets. There are maybe a few small areas where ECE3L fits a little bit better but there are other areas where it is worse. And if we look at the local amplification maps, it is even more difficult to see a clear improvement. We need also have in mind that we compare ensemble means to one realization of the reality.
Thank you for your feedback. We understand your concerns and have addressed similar points in our response to comment 6, as well as to Reviewer #2, comment 10.

To provide a clearer comparison, we replaced the original amplification maps with CDF plots ( Figs. 13e–f), which offers a more detailed assessment of the differences between the ensemble members and the reference datasets, discussed.in L388.

19. L365/ 366: "In a warmer climate, the modulating factor can either increase or decrease sea ice states …" Would it not be most probable that the effect would be small in a future warmer Arctic if we would follow your argumentation from the section before? We know from future climate simulations in the Arctic that ice thickness will further decrease and atmospheric stability decrease. What would you expect, given this knowledge, how the modulating factor would affect sea ice states in a warmer climate, increase, decrease or little effect?
We revised the text to better reflect this expectation (L478-480):
"*In a warmer climate, the modulating factor can either increase or decrease sea ice states depending on prevailing atmospheric stability and the mean sea ice thickness, making the overall effects minimal and uncertain.*"

20. L370: Please explain (or modify or delete this statement) how a parameterization in a model can "inform effective strategies for mitigating the impacts of climate change"?

We deleted this statement from the last sentence (L483). Our goal is to improve the climate models and reduce the uncertainties in climate simulations by applying new parameterization that enables us to represent sub-grid processes or quantify its relative importance. However, we acknowledge that there is still a long way to go in improving sea ice dynamics in our global model before drawing such conclusions.

21. L375: in code and data availability: what about code and data of ECE3L?

We added the information to L491.

**Technical corrections:**

L71: delete "particularly"

Done. Now in L124

L74/75: write "sea ice in the Central Arctic" or just "it in the Central Arctic" instead of only "Central Arctic".

Done. We changed it to "it in the Central Arctic" in L128.

Caption Figure 1: "Seasonal air temperature": I think you show "monthly air temperature" or maybe the "annual cycle".

Done. We changed it to "Annual cycle of".

Caption Fig S4 and S5: "under a constant": "climate" missing? ; "as in Fig. ??": replace "??"

We have addressed the issues by correcting the caption to read "under a constant forcing" and replacing 'as in Fig. ??' with 'as in Fig. 2.' Thank you for bringing these to our attention."

L395: It seems to be strange to write into the reference that "The Blue Action project receives …". Consider deleting.

We deleted it accordingly in L514.

**Reference:**

Doescher and the EC-Earth Consortium: The EC-Earth3 Earth System Model for the Climate Model Intercomparison Project 6, *Geosci. Model Dev.*, 15, 2973–3020 (2022).

Esau, I.: Amplification of turbulent exchange over wide Arctic leads: Large-eddy simulation study, J. Geophys. Res., 112, 2007.

https://dev.ec-earth.org/projects/ecearth3/wiki/Atmospheric_initial_conditions_for_climate_predictions

Keen et al.: An inter-comparison of the mass budget of the Arctic sea ice in CMIP6 models, The Cryosphere, 15, 951–982 (2021).

Lee et al. Assessment of the Pan-Arctic Accelerated Rate of Sea Ice Decline in CMIP6 Historical Simulations. J. Clim. 36, 6069-6089 (2023).

Shu et al. Assessment of sea ice extent in CMIP6 with comparison to observations and CMIP5. Geophys. Res. Lett. 47, e2020GL087965 (2020).

Sumata et al. Regime shift in Arctic Ocean sea ice thickness. Nature 615, 443–449, doi:10.1038/s41586-022-05686-x (2023)

Wunderling, N., Willeit, M., Donges, J.F. et al. Global warming due to loss of large ice masses and Arctic summer sea ice. Nat Commun 11, 5177, doi:10.1038/s41467-020-18934-3 (2020)

Citation: https://doi.org/10.5194/egusphere-2024-1865-RC2

Thank you for your thorough review and insightful comments on our manuscript titled "Modulating surface heat flux through sea ice leads improves Arctic sea ice simulations in the coupled EC-Earth3." We appreciate the opportunity to clarify our findings and address your concerns.

We have revised the manuscript and included a **marked-up version**, where removed text is shown in red and new text in blue. For your convenience, we have also updated the line numbers and new figure numbers in our previous point-by-point reply to align with the **marked-up** version.

**General:**

The manuscript aims at improving the Arctic sea ice state by introducing a modulated factor for sensible heat flux (based on Davy and Gao, 2019) depending on ice lead characteristics and the atmospheric boundary stability (enhancing in winter stable conditions and reducing it in summer unstable conditions). The scheme is shown to be effective during cold climate, although its impact seems to be less in normal or warm climate. The paper is well written. Unfortunately, some important details are missing and prevent a full acceptance of the manuscript at this stage. The heat exchange is not clearly established in view of the modulating factor and the cold and warm experiments are not clearly defined.

**Major comments:**

1.L87: "We emphasise that in ECE3L, the ocean does not supply additional heat to warm the atmosphere" (z for emphasize?) I am still scratching my head on this. However, no substantial addition is made to clarify the statement. This manuscript clearly states that the factor is applied to surface sensible heat flux: (L77-78) "we introduced a factor Alead to the surface sensible heat flux (SSHF) within the coupled ECE3 framework, to better represent the heat exchange through leads in sea ice ». Does it mean that the heat exchange between air and ocean-ice is non conservative? According to Davy and Gao (2019), only the heat flux over open water should be amplified but I was not able to ascertain the exact method used in the manuscript. Also, why stopping at sensible heat? Latent heat should be also pretty high over ice leads. Some results from Davy and Gao (2019) could be added in appendix since theirs is a project report to their funding agency (i.e., not clear whether it was peer-reviewed). Moreover, the details could be moved to the main text rather than the appendix since it is the core of the paper.

We revised it in L141. We apologize for any confusion caused by our statement. When we emphasized that in ECE3L, "the ocean does not supply additional heat to warm the atmosphere," our intent was to clarify that the heat extracted from the ocean per square meter, under the same air-sea temperature difference, remains consistent according to the bulk formula, regardless of whether it's in an open ocean grid cell or a grid cell with a small fraction

of open water surrounded by sea ice. This implies that the ocean is not more prone to freezing, as the same amount of heat is transferred to the atmosphere. The introduced modulating factor, Alead, primarily affects the efficiency of heat transfer received by the atmosphere through the "amplified/damped" surface sensible heat flux (SSHF) over leads surrounded by sea ice. Consequently, this sub-grid process is non-conservative. (As reply to Reviewer #1, minor comment 9)

Davy and Gao (2019) explained that the heat flux over sea ice leads (characterized as small open water fraction within the grid cell) should be amplified. Specifically, we implemented this as at SIC >70% in a grid cell, the SSHF calculated over the fraction of open water will be amplified by the modulation factor.

We acknowledge the importance of latent heat flux over ice leads. We  addressed this limitation in the discussion section of our revised manuscript in L418.

"*The present study builds on this foundation by performing a sensitivity analysis focused on sensible heat flux, avoiding assumptions about the latent heat flux response to leads due to the absence of data from the original LES simulations of Esau (2007).* "

It is correct that Davy and Gao (2019) is a project report and non-peer-reviewed, while it provides a valuable foundation for our study. To address this, we included key methodological details from Davy and Gao (2019) to a new Section 2.1 "*Empirical relationship for surface heat flux amplification Alead over sea ice leads*" and revised the context in Section 2.2. We incorporated relevant figures and findings from Davy and Gao (2019), particularly the results from the LES study as to how the fluxes from leads depend upon lead width under different atmospheric stabilities and how that was combined into a single amplification factor based on a lead width frequency distribution [Figures 1 and 4 from Davy and Gao (2019)], into the appendix to provide further context and support for our approach. We hope this will help clarify the methodology and ensure that the heat exchange process is more clearly understood.

2.L99 states that "The [cold and warm climate] simulations used constant forcing with a repeating seasonal cycle corresponding to the respective climate states ». However, I am still scratching my head on how you do this. In a coupled model, you have only variations is solar radiation at the top of the atmosphere and aerosol forcing. Nudging perhaps of one of the components? Please give more detail.

We revised the text in L156.

"*The coupled simulations used the CMIP6 historical external forcing from the given year (including solar radiation, GHGs concentrations, aerosols and land use, etc). The ocean and atmospheric variables still freely evolve in the simulation without being constrained.*"

3.Methodology: Some figures show the ensemble envelop (5-95% percentile), which is interesting since it gives us the statistical significance of the results. However, except for ice volume, they seem to be hardly significant (i.e., an overlap is visible), which needs to be stated in the text.

We acknowledge the importance of carefully interpreting the results. To address this, we have performed a paired *t*-test to assess statistical differences and will include a statement in the text to clarify this point. Specifically,  we conducted the paired samples *t*-test in Python with Scipy library containing the ttest_rel() function. The resulting *p*-values have been added to the figure captions for Fig.2, Fig.7 and Fig.9 , in response to minor comments 5, 6 and 7, respectively. Additionally, we have assessed the significance for Fig. 12 as well.  Finally we  included the results in lines 239, 288, 302, and 344 for the respective figures.

4.L194-195 states that "The findings suggest that during Arctic winters, the decrease in sea ice concentration is mainly due to heat advected by the ocean from the south, rather than air-sea heat exchange through sea ice leads ». I think I see the same ice reduction in S6 for concentration below 70% but I am not sure I understand the statement any better. Since the modulation factor is one over these regions, it must a non-local effect as mentioned by the authors. However, since we are dealing with a coupled system, it could be the ocean or the atmosphere. I am less inclined to think it is the ocean, as the authors do, since you would need to explain a change in the Arctic ability to pump more Atlantic waters northward, than a simple advection/diffusion of the warmer atmospheric boundary layer southward. So, please elaborate.
We acknowledge the reviewer's point.  Combining our response to Reviewer #1, comment 13, we revise this section in L262-268.
*"The findings suggest that during Arctic winters, overall sea ice thinning, particularly at the ice margins, is a key driver of sea ice concentration reduction. In these dynamic marginal zones, where the ice is often thin and fractured, even small reductions in thickness can lead to substantial decreases in ice extent. In contrast, the concentration in the central Arctic's pack ice remains close to 100% during winter, even with thickness reductions of one meter or more. This thinning at the margins coincides with a significant rise in surface air temperature by approximately 2 degree (Fig. 5), indicating a warmer atmospheric boundary layer extending southwards."*

**Minor comments:**

1.L56: Essential Climate Variables : why the capitalization here?
We have removed the capitalization of "Essential Climate Variables" in L71.

2.L248: what was the concentration threshold value used to define the ice edge? Please add.
As requested, we added "*with a 15% threshold for the sea ice edge (Goessling et al., 2016)*" to L335, though it has been originally defined in L205, Section 2.4 Validation Data and Metrics.

3.L1 of Fig1 caption: please remove "compared"
It's removed.

4.L2 of Fig1 caption: "based on" I think you meant "relative to", i.e., the bars and maps are the difference between ECE3L and ECE3 base runs (?) for the two climate experiments.
Thank you for your comment. The term "based on" has been removed.

To clarify that only the output of the ECE3 baseline simulations is used for the analysis in the figure, rather than implying a comparison of relative differences between ECE3L and ECE3 runs, the figure caption now begins with: "*ECE3 baseline simulation: (a)...*".

The figure compares surface temperature conditions between cold and warm climate states in the ECE3 model. It illustrates both the seasonal variations (Fig. 1a) and the spatial extent of sea ice cover (with SIC>70%, Figs. 1b and 1c). When (Fig. 1a) and where (Fig. 1b,c) the temperature difference is positive, an amplification factor (Alead >1) should be implemented to the ECE3L model, reflecting the reduced effect of the modulation factor in the warmer climate state.

5.Fig2: the volume does show statistical differences but not the area (at least not clearly). To be mentioned in the text.
We performed paired t-tests for Fig. 2a, b, c, and d to evaluate the statistical significance of the differences. The results are included in L239: "*The mean differences are statistically significant (p < 0.05 in a paired t-test), except for SIA in ExpWarm (p > 0.05 in Fig. 2c).*" and figure caption: "*The mean differences are statistically significant (p < 0.05, paired t-test) in panels (a), (b), and (d), but not in panel (c).*"

6.Fig.7: can the authors add the spread? I am worried that the significance is less than visually shown.
We have added the spread to Fig. 7. The ensemble means show significant differences for both Arctic sea ice area (p = 1.47 × 10$^{-10}$) and sea ice volume climatologies (p = 3.35 × 10$^{-17}$) in L288.

[Figure]

[Figure]

**Figure 7.** Comparison of annual cycle in the transient climate (1980–2014) between ECE3 (black) and ECE3L (blue) for (a) Arctic sea ice area and (b) Arctic sea ice volume. Thick lines represent the ensemble means, while the shaded areas indicate the spread between the 5th and 95th percentiles across 20 ensemble members, grey for ECE3 and light blue for ECE3L. Observations for the sea ice area include NSIDC and OSI-450a datasets, both remapped to the NSIDC-0051 grid. Sea ice volume is based on PIOMAS domain criteria (thickness > 0.15 m). The mean differences are statistically significant (p < 0.05, paired t-test).

7.Fig.9: The plots do not show a statistically different mean (the two envelops overlap).
We assessed the statistical differences. The results are included in L302. "*In Fig. 9, the September sea ice extent and the March sea ice volume show significantly different ensemble means (p < 0.05), driven mainly by external forcing. After detrending, residuals for both are no longer significantly different (p > 0.05).*" and figure caption: "*Ensemble mean differences are*

*statistically significant (p < 0.05, paired t-test); however, after detrending, residuals for both models are not significantly different (p > 0.05)."*

We corrected it as noted.

We adjusted the figure legend to Fig.11, as suggested.

[Figure]

**Figure 11.** Integrated Ice Edge Error (IIEE, defined in section 2.3) maps of ECE3L (a) and ECE3 (b) vs. NSIDC-0051 for March sea ice climatology (1980-2014). Red and blue indicate whether the model's ensemble mean overestimates or underestimates the ice edge prescribed by NSIDC-0051, respectively. (c) and (d) as in (a) and (b), but for September sea ice. Sea ice edge is defined by the 15%-sea ice concentration contour.

10. Fig.13 and L291-295: Despite the authors' vigorous statement that ECE3L Arctic temperature trend is better than ECE3, both modelled trends are still quite far from the observed one. So, we are still missing out on the amplification factor in climate simulations. I realize that it is touched upon in the previous paragraph L287-290 discussing S10 but I must say that the S10a was not showing clearly this underestimation (but it is clearer in Fig.13), unless we are talking about something subtle hidden behind the spread of the ensemble? Why would there be a

compensation between a global overestimation and the Arctic underestimation? Can you elaborate on this please, and possible in view of what the authors intent with the supplement (see below)?

We acknowledge the reviewer's point. Please refer to our response to Reviewer #1, minor 6). We deleted the original figures Fig. 13d–f, showing the local amplification ratio and we revised the corresponding texts in abstract, the methods and results discussion (L17, L216, L377-381). We added CDF plots to replace Figs. 13d–f, discussed them in L388, and emphasized the findings in the conclusion (L472–475). The new results **align with our previous findings** on the local amplification ratio but provide stronger evidence through enhanced visual and statistical assessments.

To address your concerns: Using the time series from the original Fig. S10a, we applied CDF to both visually and statistically assess the differences in variability and central tendency between the two ensembles (ECE3 and ECE3L) for the Arctic region (north of 66.5°N). As shown in Fig. 13, the Greenland-Iceland- Norwegian Seas (GIN: 40°W-15°E, 66.5°N-82°N) and the Barents and Kara Seas (BAKA: 15°E-100°E, 70°N-82°N) exhibit relatively remarkable changes in temperature trends and amplification ratio, compared to the rest of the Arctic domain. We applied CDF to the two key sub-regions.

[Figure]

**Figure 13**. Cumulative distribution function (CDF) plots for TAS trend from ECE3 and ECE3L ensembles: (d) the Greenland-Iceland- Norwegian Seas (GIN: 40°W-15°E, 66.5°N-82°N), (e) the Barents and Kara Seas (BAKA: 15°E-100°E, 70°N-82°N) and (f) the Arctic Circle. The locations of GIN and BAKA are shown in Fig. 1c.

In the Arctic, ECE3L simulates trends that are not only closer to the observed values but also with less variability compared to ECE3. This indicates that ECE3L more accurately captures the overall warming in the Arctic. In GIN, ECE3L exhibits a narrower distribution of temperature trends, demonstrating a closer alignment with the reference observational datasets. ECE3, in contrast, shows greater variability and tends to predict slightly higher warming trends than the observed references. In BAKA, ECE3L performs better than ECE3 by closely aligning with the observed warming trends. While some ECE3L estimates exceed 1.0 K/decade, it still provides a more accurate and consistent representation of temperature trends in this region compared to ECE3. Overall, these results suggest that ECE3L improves the representation of Arctic warming trends, particularly in terms of reduced variability and better alignment with observed data across key Arctic regions.

We deleted Fig.S10, because Fig.S10b shows indistinguishable differences of global mean T2m anomaly between ECE3 and ECE3L. To provide a clearer comparison, we replaced the original amplification maps with CDF plots ( Figs. 13e–f), which offers a more detailed assessment of the differences between the ensemble members and the reference datasets, discussed.in L388.

**Appendix:**
11.L480: units are missing from a1 and a2.
We added units (L652)

12.L481: is it not too large for winter Arctic Atmospheric Boundary Layer? Maybe add the expected and reasonable values for comparison.
We added an explanation for the two extreme conditions in the appendix in L653. *"This range was derived from the LES simulations, which were tested under conditions from strongly stable stratification (30.7 K km−1) to weak stratification (9.7 K km−1 ). No more strongly stable stratification was observed in the lowest model level."*

13.Supplement figures: What is the goal of them: do the authors intent to keep them there (please revise the captions then with attention), or ultimately discard them?
Thank you for the suggestion. We revised the figure captions to highlight the purpose of supporting materials and withdraw unimportant figures.

Fig. S1: We added *"The figure illustrates the modulation of turbulent heat fluxes over sea ice, showing remarkable seasonal variation and such effect in different magnitudes between ExpCold and ExpWarm scenarios, particularly in regions with sea ice thicker than 1 m."*

Fig. S2: As response to reviewer #1, major comment 2, we merged Fig.S2 and S3 to one and added the transient (r5) simulation (historical + SSP2-4.5, red) experiments from 1965-2065, to illustrate the effect of initial sea ice states and external forcing. We will add "Figure S2. *Yearly mean Arctic sea ice area (a), Arctic sea ice volume (b) and global 2-meter air temperature (c) from 1965 to 2065, comparing the ECE3 (black), ECE3L (blue), and transient (r5) simulation (historical + SSP2-4.5, red) experiments. Results from the 1985-forcing experiments are shown for the period 1985–2035, while results from the 2015-forcing experiments cover the period 2015–2065.*

*The changes in sea ice area and volume from ECE3 to ECE3L shows notable differences between the ExpCold and ExpWarm setups, revealing the sensitivity of sea ice evolution to initial seat ice conditions and external forcing, with relatively more sea ice reduction in ExpCold. The paired simulations are stable over the 50 year period in both forcing experiments, with no significant warming trends attributable to initialization artifacts. This stability is reflected in the absence of noticeable temperature drift in both forcing periods. In contrast, the transient simulation (r5), driven by historical and SSP2-4.5 external forcings, shows a clear warming trend over time. The small fluctuations in ECE3 and ECE3L are consistent with internal variability, indicating that the model does not exhibit any pronounced initialization-induced warming."*

Figure S3: We added *"In summer the regions experiencing the greatest SIT reduction remain consistent with those identified in the winter, as shown in Fig. 3: from the eastern Arctic in the*

*thicker ice regions in ExpCold shifting to the western Arctic in the thinner ice regions in ExpWarm.”*

Figure S4: We added *“In summer the regions experiencing the greatest SIC reduction remain consistent with those identified in the winter, as shown in Fig. 4. These are primarily in the thinner ice regions, typically along the ice margins, with up to 30% reduction in ExpCold compared to less than 20% in ExpWarm.”*

Figure S5: As reply to Reviewer #1, major 3 & minor 11, we showed the ensemble mean and the spread of TAS and AMOC in ECE3L, compared with those in the original ECE3 ensemble.

We added *“For the generation of the ECE3L ensemble, we introduced small random perturbations (on the order of $10^{-5}$ °C) in the 3D temperature field. Although these perturbations are minor, they are adequate to induce divergence among ensemble members after only a few days. We compared the ensemble mean and the model spread for TAS (a), and AMOC (b) between ECE3L and the original ECE3 ensemble. We applied paired t-test to assess statistical differences over the period from 1980 to 2014. Although the difference in AMOC between r5 and r8 has converged since 1980, the ECE3L simulations exhibit a model spread (std=1.4 Sv) similar to that of ECE3 (std=1.5 Sv), with an ensemble mean difference of -0.3 Sv (ECE3 - ECE3L). This difference is statistically significant ($p < 0.05$), indicating that our method captures the internal variability of AMOC adequately. Similar to the global mean T2m, both simulations exhibit a comparable model spread of 0.2 °C, with an ensemble mean difference of -0.1 °C. This difference between the ensemble mean time series is statistically significant ($p < 0.05$). However, the detrended time series do not show a significant difference ($p > 0.05$), suggesting that external forcing plays a dominant role in the observed variations.”*

Figure S6: We added *“The regions showing the greatest SIC reduction in March and September climatologies are similar to those observed in ExpCold (Figs. 4a,b and S4a,b), though the magnitude of the reduction is lower compared to ExpCold.”*

Figure S7: We removed this figure, as the comparison with OSI-450a observations leads to the same conclusion as Fig. 11.

Figure S8: We removed this figure for the same reason as Figure S7.

Figure S9: We removed this figure for the monthly comparison of IIEE time series, as ECE3L consistently shows lower errors in sea ice edge representation across all seasons. We think Fig.12 is sufficient to present the IIEE for September and March, highlighting the errors during the sea ice minimum and maximum conditions.

Figure S10: We removed this figure as reply to comment 10.

**Reference:**

Davy, R. and Gao, Y.: Improved key process in representing Arctic warming (D3.5), https://doi.org/10.5281/zenodo.3559470, The Blue-Action project (the European Union's Horizon 2020) Research and Innovation Programme under Grant Agreement No.727852., 2019.

Goessling et al.: Predictability of the Arctic sea ice edge, Geophys. Res. Lett., 43,1642–1650, 2016.

Hersbach et al.: The ERA5 global reanalysis, Quart. J. Roy. Meteorol. Soc., 146, 1999–2049, 2020.

Kanamitsu et al.: NCEP-DOE AMIP-II Reanalysis (R-2), Bull. Amer. Meteor. Soc., 83, 1631–1644, 2002.

Kobayashi et al.: The JRA-55 reanalysis: General specifications and basic characteristics, J. Meteor. Soc. Japan, 93, 5–48, 2015.

Rohde and Hausfather: The Berkeley Earth land/ocean temperature record, Earth System Science Data, 12, 3469–3479, 2020.

---

## Referee Report (RR1)

**Review**

"Impact of modulating surface heat flux through sea ice leads on Arctic sea ice in EC-Earth3 in different climates" by T. Tian, R. Davy, L. Ponsoni, S. Yang

**General:**

I appreciate the additional analysis and modification the authors have made and thank for the detailed explanations and clarification as response to my questions and suggestions.

The article has been substantially improved, and to me the manuscript is almost ready for publication. I have only a very few minor comments:

**My old comment 6**/ new Figure 13 with respect to improvements of trends: Thank you for showing the CDFs for the temperature trends in the three regions. Maybe I do not understand this, but I do not really see why you would judge from these figure that the trends are showing an improvement in ECE3L compared to ECE3?  I see a reduced spread in the trends across the members in ECE3L compared to ECE3. While it could be possible that the reduced sea ice bias in ECE3L leads to a reduced and potentially more realistic spread in the trend across members, we do not know if that is really an improvement because we do not know how large the spread in reality would be if we would have an 'ensemble of realities'. I do not think that we can judge from the fact that more ensemble members are closer to the observed trend that this is more realistic. Maybe the reality itself was not a central value but an outlier?

**My old comment 7:** Sorry to insist a bit on this point: Why should any coupled model reproduce the large observed ice decrease between 2000 and 2012? The observed reduction was clearly due to a combination of decreasing trend and ('negative') internal variability, similar as the missing trend after 2012 is also a combination of negative trend and ('positive') variability. It would be different if CMIP6 models would not at all be able to simulate sea ice reduction periods of the observed magnitude but I do not think this is the case.

**Line 9:** the spatial pattern is similar between cold and transient run but the amplitude is different, right? To clarify this, I would suggest to write "spatial patterns of the mean sea ice changes in …" instead of "spatial changes". I would suggest the same change in the conclusions.

I am still surprised that your 1985-control run is hardly warming and sea ice volume even slightly increasing after initializing from year 1985 of the transient historical run. Normally, you would assume that net radiation is not in balance in a transient run and you would see some warming thereafter. However, also in your 2015-year control run, no additional warming seems to happen, and I agree that your results suggest that this seems not to be an artifact of internal variability.

**Line 13:** replace "overestimated" with "overestimation" ?

**Line 66:** Both in the title and in the first line of section 2.1. If you mention "relationship", I would expect a "relationship between A and B",  but you are only mentioning one thing (in the title) and nothing (in the first sentence).

---

## Author Response (AR2)

Reviewer #1:

General:

I appreciate the additional analysis and modification the authors have made and thank for the detailed explanations and clarification as response to my questions and suggestions. The article has been substantially improved, and to me the manuscript is almost ready for publication. I have only a very few minor comments:

We appreciate the reviewer's thoughtful feedback and efforts in improving this manuscript. We have addressed the minor comments and incorporated the suggested revisions, highlighting the changes in bold while keeping the original text in italic for reference.

We have revised the manuscript and included a marked-up version, where removed text is shown in red and new text in blue. For your convenience, we have also updated the line numbers and new figure numbers to align with the marked-up version.

1. My old comment 6/ new Figure 13 with respect to improvements of trends: Thank you for showing the CDFs for the temperature trends in the three regions. Maybe I do not understand this, but I do not really see why you would judge from these figure that the trends are showing an improvement in ECE3L compared to ECE3? I see a reduced spread in the trends across the members in ECE3L compared to ECE3. While it could be possible that the reduced sea ice bias in ECE3L leads to a reduced and potentially more realistic spread in the trend across members, we do not know if that is really an improvement because we do not know how large the spread in reality would be if we would have an 'ensemble of realities'. I do not think that we can judge from the fact that more ensemble members are closer to the observed trend that this is more realistic. Maybe the reality itself was not a central value but an outlier?

We acknowledge with the reviewer that Fig. 13 (now is Fig. 12) shows better alignment in trend with the observation that is the only realisation. We replace the term 'improvement' with more neutral wording in Section 5.1, which discusses how sea ice evolution influences Arctic warming. We made minor adjustments in the abstract and conclusion.

*Abstract: Notably, ECE3L* **shows closer** *alignment with observational data and refines the declining sea ice volume trend overestima**tion** in ECE3, reducing **overestimated** ensemble variability **caused by excessive sea ice. This, in turn, amplifies sea ice** sensitivity to Arctic warming, particularly in the marginal ice zone. These findings **emphasize** the importance of **accurately** representing surface heat flux through sea ice leads, which plays a critical role in capturing the influence of atmospheric stability on sea ice dynamics and regional Arctic amplification.*

*L344: This study provides valuable insights for Arctic climate modelling. By incorporating a modulating factor for surface sensible heat flux over sea ice to **account for** processes over leads, the EC-Earth3 model shows **closer agreement with observed** Arctic sea ice extent and volume, particularly under colder climate conditions. This adjustment **mitigates** a known bias in earlier simulations.*

*L351: The **parameterisation introduced** in this study supports these insights, **emphasizing the need to represent** finer-scale ocean-sea ice-atmosphere coupling processes.*

*L361:  The temperature trend maps for 1980–2014 (Fig. **12**a-c) **show** that ECE3L **more closely aligns with observed trends along the ice edge in the North Atlantic sector of the Arctic compared to ECE3, which** overestimates the warming trend in the Barents Sea**, while underestimating it in** the Greenland and Labrador seas. Additionally, ECE3L **represents** the warming trend **in** the East Siberian Sea, unlike ECE3, which consistently underestimates the trend in the **Pacific sector of the** Arctic.*

*L367: The cumulative distribution function (CDF) analysis in Fig. 13d-f emphasizes regional differences in ensemble performance, with ECE3L **exhibiting** reduced variability and a **central tendency that aligns more closely with observations than** ECE3. This **results in more consistent representation of** warming trends across the Greenland-Iceland-Norwegian Seas (GIN), the Barents and Kara Seas (BAKA), and the broader Arctic. Consequently, ECE3L **achieves closer alignment with observed** local amplification ratios (the ratio of local warming to global mean warming; Rantanen et al., 2022), **leading to more confined** estimates of Arctic amplification.*

*L444: ECE3L **shows reduced ensemble variability**, **leading to enhanced sea ice** sensitivity to Arctic warming and **providing more constrained** estimates of Arctic amplification.*

2. My old comment 7: Sorry to insist a bit on this point: Why should any coupled model reproduce the large observed ice decrease between 2000 and 2012? The observed reduction was clearly due to a combination of decreasing trend and ('negative') internal variability, similar as the missing trend after 2012 is also a combination of negative trend and ('positive') variability. It would be different if CMIP6 models would not at all be able to simulate sea ice reduction periods of the observed magnitude but I do not think this is the case.

We agree that the decadal variability may have played an important role in the observed declining trend between 2000 and 2012. Our intention was to highlight the risk that, while the multi-model mean can capture periods of rapid sea ice loss, models with excessive sea ice volume may alter their sensitivity to external forcing. We have revised it

L30: "*The CMIP6 multi-model mean captures the observed **decline in Arctic sea ice in general, while substantial variability exists both among different models and among ensemble members of the same model (Lee et al., 2023), highlighting internal variability as a key source of uncertainty in decadal trends (Dörr et al., 2023). A major challenge lies in ice thickness representation—models with thicker ice tend to exhibit a faster decline** in sea ice volume **than those with thinner ice, increasing uncertainty in reproducing the overall rate of decline (Lee et al., 2023; Massonnet et al., 2018). Ice thickness also influences** feedback mechanisms, such as the ice-albedo effect**, where thinner ice and earlier melt expose open water, accelerating warming (Bhatt et al., 2014). Additionally, missing processes like surface heat flux over leads — open water areas within sea ice cover, which mediate ocean-atmosphere heat exchange in winter, can amplify local warming (Esau, 2007; Marcq and Weiss, 2012). These factors*

***contribute to uncertainties in modelling Arctic warming, projecting** future sea ice evolution, **and climate impacts (Wunderling et al., 2020).***"

3. Line 9: the spatial pattern is similar between cold and transient run but the amplitude is different, right? To clarify this, I would suggest to write "spatial patterns of the mean sea ice changes in …" instead of "spatial changes". I would suggest the same change in the conclusions.

I am still surprised that your 1985-control run is hardly warming and sea ice volume even slightly increasing after initializing from year 1985 of the transient historical run. Normally, you would assume that net radiation is not in balance in a transient run and you would see some warming thereafter. However, also in your 2015-year control run, no additional warming seems to happen, and I agree that your results suggest that this seems not to be an artifact of internal variability.

The revised phrase in blue with the original in black and italic:

Abstract: L9

"*The **spatial patterns of mean** sea ice **changes** in the transient climate closely resemble those observed in the cold-climate experiment*."

Conclusion: L438

"*The spatial **patterns of mean sea ice changes** from 1980 to 2014 in ECE3L closely mirror those simulated using 1985-forcing (cold-climate)*."

4. Line 13: replace "overestimated" with "overestimation" ?
Done.

5. Line 66: Both in the title and in the first line of section 2.1. If you mention "relationship", I would expect a "relationship between A and B", but you are only mentioning one thing (in the title) and nothing (in the first sentence).

Thank you for the suggestion. We revised accordingly:

L74 Title: "*2.1 Empirical relationship **between** surface heat flux amplification $A_{lead}$ **and** sea ice leads*"

L75: "*An empirical parameterisation, introduced by Davy and Gao (2019) for the NorESM model, defines **the relationship between surface sensible heat flux (SSHF) amplification and sea ice leads***."

**References:**

Bhatt, U. S. et al. Implications of Arctic sea ice decline for the Earth system, Annual Review of Environment and Resources, 39, 57–89 (2014).

Davy, R. and Gao, Y. Improved key process in representing Arctic warming (D3.5), https://doi.org/10.5281/zenodo.3559470 (2019).

Deser, C. et al. The seasonal atmospheric response to projected Arctic sea ice loss in the late twenty-first century, J. Climate, 23, 333–351 ( 2010).

Docquier & Koenigk. Observation-based selection of climate models projects Arctic ice-free summers around 2035. Commun. Earth Environ., 2, 144 (2021).

Doescher, R. & the EC-Earth Consortium. The EC-Earth3 Earth System Model for the Climate Model Inter-comparison Project 6, Geosci. Model Dev., 15, 2973–3020 (2022).

Dörr, J. S. et al. Forced and internal components of observed Arctic sea-ice changes, The Cryosphere, 17, 4133–4153 (2023).

Esau, I. Amplification of turbulent exchange over wide Arctic leads: Large-eddy simulation study, J. Geophys. Res., 112 (2007).

Keen, A. et al. An inter-comparison of the mass budget of the Arctic sea ice in CMIP6 models, The Cryosphere, 15, 951–982 (2021).

Lee, Y. J. et al. Assessment of the Pan-Arctic accelerated rate of sea ice decline in CMIP6 historical simulations, J. Climate, 36, 6069–6089 (2023).

Marcq, S. and Weiss, J.: Influence of sea ice lead-width distribution on turbulent heat transfer between the ocean and the atmosphere, The Cryosphere, 6, 143–156 (2012).

Massonnet, F. et al. Arctic sea-ice change tied to its mean state through thermodynamic processes, Nat. Clim. Change, 8, 599–603 (2018).

Notz and SIMIP Community. Arctic sea ice in CMIP6. Geophys Res Lett 47, e2019GL086749 (2020).

Wunderling, N. et al. Global warming due to loss of large ice masses and Arctic summer sea ice, Nat. Commun., 11, 5177 (2020).

Reviewer #2:

The authors responded to the different points raised by the reviewers in a satisfactory manner… with a few exceptions. I still have some remaining points that require discussions which pushes me to still upheld any decision to accept the paper at this stage:

We appreciate the reviewer's feedback on this manuscript. We have addressed the comments point by point and incorporated the suggested revisions, highlighting the changes in bold while keeping the original text in italic for reference.

Additionally, we have provided a marked-up version of the manuscript, where deleted text is shown in red and new text in blue. To ensure clarity, we have updated the line numbers and figure numbers accordingly to align with this version.

1. implementation for the heat flux exchange: the authors in their response explicitly state that the modulation of SSHF is non-conservative. However, the language used is not totally clear to me. Just to make sure that we mean the same thing: the ocean over open leads of relative area A feels a flux F so that A.F is the total flux for that entire grid cell (assuming zero flux under sea ice) but the atmosphere receives a.A.F, i.e., where a is the additional modulation flux? Understandably, this non-conservative aspect worries me since it is not physical. Is there a particular reason for not conserving the flux exchange between the two earth components? [Maybe the authors need to rephrase the sentence on Line 124: "This means the ocean is not more prone to freezing, as the same amount of heat loss to the atmosphere » as the subordinate (second part) is missing a verb.] If my interpretation is correct, it would mean that the sea ice thickness and cover is reduced in winter because the amplified flux warms the atmosphere which feeds back to the sea ice as a warmer boundary condition. However, this precludes any interaction with the ocean which, on the other hand, should be cooler (due to the amplified flux lost to the atmosphere) and would form more ice. Hopefully, this is just a communication issue.

Thank you for your comment. The non-conservative phrasing in the original text resulted from a misinterpretation, as the role of the coupler was overlooked. The coupled ECE3 framework operates in the following order:

1) The atmosphere computes all fluxes (including SSHF) using bulk formulas.
2) The ocean/sea ice components receive these fluxes directly via OASIS3-MCT, meaning the same flux (e.g. $A \cdot F$) is used consistently across components (Döscher et al.2022, Table 3 in Section 2.2).
3) In ECE3L, Alead modifies the atmospheric sensible heat flux (e.g., amplifying it to $a \cdot A \cdot F$), the adjusted fluxes warm the atmospheric surface, which may enhance ice melting in the winter. Since the ocean model does not compute the flux independently but instead receives it from the atmosphere via the coupler, the process remains conservative in principle.

The previous description was misleading because we emphasized that the ocean fluxes calculated from bulk formulas with no changes. Then the implementation of Alead emphasized a "one-way" modification to the atmosphere model, which was true only for our atmosphere-only experiment.

However, in the coupled experiments, the OASIS-MCT coupler ensures that the ocean receives the fluxes as determined by the atmosphere. The input files to the coupler, providing the ocean surface state variables (incl. sea surface temperature, sea ice concentration, sea ice temperature) to the atmosphere for computing surface fluxes, and remapped those fluxes to the ocean/sea ice model grid via the coupler. Importantly, the coupling time step between the atmospheric, ocean, and sea ice components is

synchronized at 2700s (Table 2 of Döscher et al. 2022), meaning that the fluxes exchanged between these components remain dynamically consistent.

L129: We have revised the text by clarifying the role of OASIS3-MCT. "*In ECE3L, **all surface fluxes are computed in the atmosphere using state variables from the ocean-atmosphere interface and then remapped to the ocean and sea ice components via the OASIS3-MCT coupler (Doescher et al., 2022).** The modulating factor, Alead, influences only the sensible heat flux in the surface atmosphere by either "amplifying or damping it over leads, depending on atmospheric stability. This adjustment can increase up to 1.2, enhancing surface heat exchange over sea ice where SIC exceeds 70%, particularly during winter. Conversely, in summer, the factor decreases from 1 to 0.9, leading to a reduction in surface heat exchange and producing the opposite effect on the surface atmosphere (see Fig. S1 and Section 3.2).*"

We agree with your interpretation: While amplified winter heat loss (Alead) enhances localized basal freezing, the model also captures central Arctic thinning due to a warmer atmospheric boundary layer. Since thinner ice is more vulnerable to melting, this results in reduced ice extent along the ice edge, despite the modulation factor being restricted to the central pack. However, we have decided to remove this aspect, as basal freezing is outside the scope of our study.

2. The second concern is the reduced spread of the ECE3L transient run. The authors did show in their response and revised manuscript that the reduced variability (e.g. new Figure 13d-c), is not due to the peculiarities of the initialization from two perturbed members since the AMOC (arguably the largest source of internal variability in the system) spread is recovered after a short initial spinup. The authors conclude that the reduced spread is beneficial as it reduces the error relative to the available datasets. However, I don't think the source of the reduced spread/variability is discussed. I personally speculate that the modulation factor acts as a negative feedback mechanism, warming the atmosphere in cold climates and cooling it in warm climates, thus reducing the intrinsic variability of the Arctic climate. This, of course, relates partly to my concern about unphysical flux exchanges in the previous point.

Thank you for the suggestion. We argue that the role of negative feedback is not evident in this context. As shown in Fig. 2, *the interannual variability in ECE3L closely aligns with that of ECE3 across all months in both ExpCold and ExpWarm. This indicates that the parameterization does not alter the system's internal variability.* We have added this to L226-228.

In transient simulations, the smaller model variability in ECE3L is accompanied by substantial reductions in overestimated sea ice area and volume prior to 1990 (L292–300). This suggests that the narrower spread resulted from improved sea ice representation in simulations (prior to 1990) with excessive sea ice, without significant changes to other simulations or periods.

We propose addition discussion in the next paragraph (L301) as follows:

***In this sensitivity study, the parameterisation amplifies winter heat loss to reduce ice thickness and dampens summer heat uptake to delay melt, potentially addressing seasonal biases such as excessive winter ice thickness and premature summer melting, which would otherwise shift the annual minimum from September to August (Döscher et al., 2022; Keen et al., 2021). In ECE3, Alead is most effective in colder conditions with excessive sea ice, where greater sea ice coverage and atmospheric stability contribute to large model variability before 1990 (Fig. 7). However,** as the Arctic warms**, its influence weakens due to reduced** winter stratification **and continued summer sea ice retreat (Deser et al., 2010). Consequently, its impact on mitigating summer sea ice bias remains limited. Thus, the smaller spread in ECE3L is a direct***

*result of bias reduction in sea ice representation rather than an artificial constraint on variability."*

3. Another point related to the reduced spread is the impact on the scores. The authors state regularly that they favor their ECE3L setup which has a smaller spread and better scores, seemingly implying that spread is "bad". This is of course quite a simplification since it all depends of the dispersive nature of the climate run. For instance, the authors have not demonstrated that the default run is too dispersive in the Arctic, just that the mean ensemble is a bit off relative to the observations. Moreover, please check Section 3 of Peterson et al. (2022) for an explanation why the scores are automatically improved due to a reduced spread. Please elaborate.

Thank you for the suggestion. We acknowledge the reviewer's concern regarding the interpretation of ensemble spread and its impact on scores. Our intent is not to imply that a reduced spread is inherently superior or that spread is "bad." Rather, the reduced spread in ECE3L results from addressing excessive sea ice through the parameterisation, which examines the impact of missing heat flux representation over leads in ECE3.

Regarding "the default run is too dispersive in the Arctic," Figures 6 and 7 demonstrate that ECE3 persistently overestimates Arctic sea ice area and volume, with observations often aligning with the lower bound of the ensemble spread. This suggests that the wide spread reflects systematic biases rather than an appropriate representation of natural variability.

The reduced spread in ECE3L arises from addressing excessive sea ice in colder conditions before 1990, which shifts the ensemble mean closer to observations. While Peterson et al. (2022) note that reduced spread can improve scores, our focus is not on tuning for better scores but on understanding how heat flux representation affects Arctic climate simulations. The narrower spread reflects the effects of this sensitivity experiment, not deliberate tuning.

We add the following to section 5.1 in L374: "*Constraining ensemble variability, which may incidentally improve certain forecast scores (Peterson et al., 2022), is not our objective. Instead, we aim to assess how the missing representation of heat flux over leads influences Arctic climate simulations. The narrower spread in ECE3L arises from the modulating effect of Alead on surface heat exchange, rather than from deliberate tuning. This underscores the sensitivity of sea ice states to subgrid-scale heat flux processes, highlighting the need for further investigation.*"

4. The results shown in Figure S3c are worrying me: they show that, in September, the modulation is thickening the ice in the Pacific marginal sea sector and decreasing in the central Arctic to the point where the former regions are thicker than the latter, in particular North of the CAA and Greenland where we typically expect to find the thickest ice. I fear here that the flux modulation is reducing too much the melt in the marginal ice zones. Please elaborate.

Thank you for your observation. However, the pattern you noted—thicker ice in the Pacific marginal sea sector and thinner ice north of the CAA and Greenland—is a known bias in EC-Earth3 historical ensembles, as documented in Doescher et al. (2022, Fig. 13): "

1) *In September, the Arctic sea ice is clearly too thick in the model, with a bias of up to 2 m compared to PIOMAS.*
2) *In March, EC-Earth3 overestimated sea ice thickness in the central Arctic but underestimated it in the Bering and Kara Seas.*
3) *PIOMAS appears to overestimate thin ice thickness and underestimate thick ice, partly explaining the higher modeled thickness in the central Arctic.*"

[Figure]

Figure 13 of Doescher et al. (2022). "*Difference in Arctic sea ice thickness between the ensemble mean of EC-Earth3 and PIOMAS in September (a) and March (b), averaged over 1980–2010.*"

We include ECE3 (e) below in addition to Fig. S3c, d for ExpWarm. The thicker ice in the Pacific marginal seas compared to north of Greenland is present in both ECE3L and ECE3L (Fig. S3c,e). The difference between ECE3L and ECE3 (Fig.S3d) indicates that ECE3L either reduces or slightly increases ice thickness in the Pacific marginal sea sector while slightly increasing it north of the CAA and Greenland. We argue that the modulating factor does not directly cause reduced ice melting in the Pacific marginal ice sea sector, because the parameterisation is confined to the central Arctic, where SIC > 70%, as shown in Fig. 1 and Fig. S1.

[Figure]

Figure S3. Late summer (September) Arctic sea ice thickness under constant forcing: for ExpWarm (c) ECE3L, (d) Difference (ECE3L-ECE3), and (e) ECE3.

To improve clarity, we revise ==L249==: "*In Arctic summer (Alead<1),* **sea ice thinning patterns** *remain consistent with* **winter** *(Fig. S3),* **indicating the dominant role of winter amplification. The thicker ice in the Pacific marginal seas compared to north of Greenland is present in both ECE3 and ECE3L and results from a known EC-Earth3 bias (Doescher et al., 2022), not the modulation effect (Fig. S3d), which applies only in the central Arctic (Fig. 1).** "

Other minor points:

5. Line 58: "introducing the modulate factor to a coupled climate model", "implementing the modulation factor into a coupled climate model" instead?

Done (Now in ==L66==).

6. Line 80 and more: I fear that there is now a text duplication between the appendix and the main body.

Thank you for your feedback. Considering the length of the main text, we have simplified Section 2.1 and included a cross-reference to Appendix A in ==L106==: "*A full derivation of this*

*parameterisation, including governing equations and parameter choices, is provided in Appendix A.*" We also removed the redundant introductory sentences from Appendix A.

7. Line 86: I don't think that the variable theta in Greek symbol has been introduced at this stage. In fact, I see it declared at line 94 instead.

Thank you for pointing this out. We have revised the text to introduce θ at its first mention (in L100).

8. Sorry, I don't understand Line 214-215. Maybe it is meant as a relation between thickness and impact of the modulation? Would a scatterplot help then?

Thank you for pointing this out. We add a scatterplot to Fig. S2d, and revise the two last sentences to improve clarity in L222: "*In ExpCold, **the modulation of heat fluxes in ECE3L consistently reduces SIA and SIV after the spin-up, leading to thinner ice compared to ECE3. In ExpWarm, however, ECE3L exhibits both increases and decreases in sea ice with minimal impact on overall thickness** (see full time series in Fig. S2).*"

[Figure]

*Fig.S2 (d) Scatter plot comparing the 30-year mean sea ice thickness (SIV/SIA, in meters) between ECE3 and ECE3L simulations. The diagonal line represents a 1:1 ratio, where points above indicate thicker ice in ECE3L and points below indicate thinner ice. Colors differentiate between the 30-year ExpCold (blue) and ExpWarm (red) periods, while the 20-year spin-up phase is shown in gray. Dashed lines indicate 30-year means for each dataset.*

9. Line 225: beware that expression like eastern/western Arctic are country centric (maybe North-American in your case?). Better use the Atlantic/Pacific sectors as done later.

L237: *In ExpCold (**Fig. 3b**), it reaches a maximum of -1 m in the central Arctic, **particularly in the region from north of Greenland to the Beaufort Sea**. In ExpWarm (Fig.3d), the largest reduction (-0.5 m) **shifts to the central Arctic, north of the Laptev and East Siberian Seas.***

10. Since I see the Supplement figures being cited often, I believe this could justify their inclusion in the main manuscript. The authors should decide what other figures to drop to limit the total number of figures. Role of the supplement: maybe the editor could enlight me since this minor point was already raised?

Thank you for your suggestion. However, we prefer to maintain the current structure, with the main manuscript focusing on key experimental results and the supplementary material providing supporting details.

For example, Figure S1 illustrates the modulation factor in ExpCold and ExpWarm, based on lapse rate and sea ice concentration from previous atmosphere-only (AGCM) simulations using CMIP6 historical forcing and surface boundary conditions from the UK Met Office Hadley Centre SST and SIC dataset (1/4-degree, Version 2.2.0.0).

Figures S3 and S4 present summer results, where the modulation effect is minimal due to the factor being below 1, sea ice extent retreat (SIC > 70%), and warm air advection from lower latitudes.

Figure S6 is omitted from the main manuscript as the spatial patterns of mean sea ice concentration changes in the transient climate remain consistent with those observed in the ExpCold (Figs. 4a,b for March & Figs. S4a,b for September).

After careful evaluation, we think including these figures in the main manuscript would not provide additional insights to readers. We appreciate your feedback and welcome any further guidance from the editor regarding the role of the supplementary material.

**11. Line 295-296: I assume you mean that the Alead < 1 in summer. You could be more explicit about it.**

Thank you for your comment. In the context, the statement in Lines 295-296 (now in ==Lines 316-318==) is not related to Alead <1 in summer, as it pertains only to ECE3 when comparing mean SIT between ExpCold and the transient climate.

This statement was originally in response to Reviewer #1's previous minor comment 5, which questioned why the impact of modulated heat flux is smaller in the transient run despite similar mean climate states in ExpCold and the transient climate.

For reference, we quoted our previous response here:

1) ECE3's mean SIT in March is slightly higher in the central Arctic for ExpCold than in the transient run (Figs. 4b & 5b), with minimal differences in summer (Figs. 4d & 5d).

2) The smaller reduction in sea ice during the transient run leads to thicker ice remaining in ECE3L (Fig. 10a).

These points directly support the statement in Lines ==318-319==, explaining why ice is generally thicker in ECE3L under the transient-climate due to the diminished effect of heat flux amplification on ice thinning compared to ExpCold.

[Figure]

Fig.4b,d                                    Fig.5b,d

Figure 4. ECE3 30-year mean sea ice thickness in March (b) and September (d) for the cold-climate experiments. Figure 5. As Fig.4, but showing the 35-year mean for the transient-climate experiments.

Thank you for your comment. By 'model-form uncertainty,' we refer to uncertainties arising from the structural assumptions and parameterizations used in models. We now replace "model-form uncertainty" with "*structural uncertainties in models*" (L394).

13. Not sure that Fig.1 is that useful. I fear also that the sign of the temperature difference is reversed between panel a and b-c.

The sign of the temperature difference in Fig. 1 aligns with Deser et al. (2010), which we followed to illustrate the temperature inversion over Arctic sea ice. The temperature difference (T850 - T100) is typically positive in winter, indicating a strong inversion where cold air near the surface suppresses vertical mixing, leading to a highly stratified atmosphere. Unlike the open ocean, sea ice limits heat exchange between the ocean and atmosphere, preventing warm oceanic influence from mixing upwards. This reinforces surface cooling and sustains the inversion.

In Fig.1, the comparison between ExpCold (dark blue) and ExpWarm (light gray) highlights the weakened inversion in ExpWarm, demonstrating the reduced role of sea ice in maintaining the inversion.

[Figure]

In Deser et al (2010) "*Left: Fig 7. Geographical distributions of the strength of the December low-level inversion (T850hPa-T1000hPa) during 1980–99 and 2080–99. Right: FIG. 8. Seasonal cycles of T850hPa-T1000hPa during 1980–99 (dark colors) and 2080–99 (light colors) over the (top) Arctic Ocean (blue) and (bottom) high-latitude continents (red).*"

14. Paired t-test or two-sided t-test?

It is a two-sided t-test. We clarify it throughout the text (L220, Captions of Figs. 2,5,6,8,11).

15. Fig.6 is of less interest to me, could be a candidate for exclusion if space is an issue.

Thank you for the suggestion. We remove Fig.6 as well as Lines 264-266.

16. Sep minimum is not recovered by the new scheme as noted by Rev #1, should be noted as part of the limitation, along with my concern in point 1.

This is addressed in response to the major comment 2. We also briefly discussed it in Section 5.2 Advances and limitations (L413): "*Moreover, the new scheme does not fully recover the observed September minimum, underscoring persistent challenges in simulating late-summer sea ice loss. This limitation stems from its focus on atmosphere-ice heat flux modification, without directly addressing ocean-ice dynamics, which are crucial for accurately capturing summer retreat (Docquier and Koenigk, 2021).*"

17. Caption of Fig. 10: "The areas with SIC≥70% in ECE3L is compassed by red lines.": I think you mean "both ECE3 and ECE3L are encompassed" here.

Yes, we have added "*in both ECE3 and ECE3L are*" the captions for clarity.

18. Fig A.1a: not sure why the climate model would show any variation with lead size! [this was not in the Esau paper]

The black line was not in the Esau paper, therefore, we removed it from Fig. A.1a. This black line represented a 2D turbulence structure, contrasting with the 3D turbulence parameterisation described in Esau (2007). It was used to illustrate general turbulence parameterisation in climate models (Davy and Gao, 2019); however, since it was not explicitly mentioned in the description here, we decided to remove it to avoid confusion.

Additionally, the red solid line has been changed to a red dashed line to improve accessibility for readers with color vision deficiencies.

19. Caption of Figure S1: "Factor modulated to turbulent heat fluxes" do you mean "Figure S1. Factor modulating the turbulent heat fluxes »?

Yes, we have revised the caption accordingly for clarity.

**Reference:**

Deser, C. et al. The seasonal atmospheric response to projected Arctic sea ice loss in the late twenty-first century, J. Climate, 23, 333–351 (2010).

Docquier & Koenigk. Observation-based selection of climate models projects Arctic ice-free summers around 2035. Commun. Earth Environ., 2, 144 (2021).

Doescher, R., Acosta, M., Alessandri, A., and the EC-Earth Consortium: The EC-Earth3 Earth System Model for the Climate Model Intercomparison Project 6, Geosci. Model Dev., 15, 2973–3020 (2022).

Peterson, K. A. et al. Understanding sources of Northern Hemisphere uncertainty and forecast error in a medium‑range coupled ensemble sea‑ice prediction system. Quarterly Journal of the Royal Meteorological Society, 148 (747), 2877-2902 (2022).